# Stop Guessing: Choosing the Optimization-Consistent Uncertainty Measurement for Evidential Deep Learning

**Linye Li[1]   Yufei Chen[1]\*   Xiaodong Yue[2]   Xujing Zhou[2]   Qunjie Chen[1]**
[1]School of Computer Science and Technology, Tongji University
[2]Artificial Intelligence Institute, Shanghai University

{linyeli, yufeichen}@tongji.edu.cn, yswantfly@shu.edu.cn
 muxixi@shu.edu.cn, 2250115@tongji.edu.cn

## Abstract

Evidential Deep Learning (EDL) has emerged as a promising framework for uncertainty estimation in classification tasks by modeling predictive uncertainty with a Dirichlet prior. Despite its empirical success, prior work has primarily focused on the probabilistic properties of the Dirichlet distribution, leaving the role of optimization dynamics during training underexplored. In this paper, we revisit EDL through the lens of optimization and establish a non-trivial connection: minimizing the expected cross-entropy loss over the Dirichlet prior implicitly encourages solutions akin to multi-class Support Vector Machines, maximizing decision margins. Motivated by this observation, we introduce the *optimization-consistency principle*, which deems an uncertainty measure valid if its value decreases as samples approach the global optimum of the training objective. This principle provides a new criterion for evaluating and designing uncertainty measures that are consistent with the optimization dynamics. Building on this foundation, we further propose a novel measure, *Margin-aware Predictive Uncertainty (MPU)*, which directly captures the separation between target and non-target evidence. Extensive experiments on out-of-distribution detection and classification-with-rejection benchmarks demonstrate the effectiveness of our propositions. [1]

## 1 Introduction

In recent years, Evidential Deep Learning (EDL) (Sensoy et al., 2018; Malinin & Gales, 2018; Charpentier & Günnemann, 2020) has emerged as a popular approach for classification tasks. By training a single deterministic network to predict the parameters of a Dirichlet distribution, EDL provides an elegant way to jointly capture aleatoric (data) and epistemic (model) uncertainty. Unlike methods that draw inspiration from Bayesian Neural Networks, such as MC-Dropout (Gal & Ghahramani, 2016) and Deep Ensembles (Rahaman et al., 2021), which require multiple models or repeated sampling to approximate posterior distributions, EDL achieves efficiency with only a single model and a single forward pass at test time. This computational advantage, together with its strong empirical performance, has led to its widespread adoption in various downstream tasks like open-set recognition (Bao et al., 2021; Zhao et al., 2023), trusted multi-view classification (Han et al., 2021; Liu et al., 2022; 2025a;b;c) and out-of-distribution detection (Gao et al., 2025; Chen et al., 2023a; Li et al., 2025; Fu et al., 2026; Qu et al., 2024).

Although EDL has been extensively studied, prior work has almost exclusively analyzed it from a probabilistic perspective. For example, by designing priors, leveraging Fisher information, or imposing constraints on Shannon entropy to improve performance (Bengs et al., 2022; 2023; Wimmer et al., 2023; Juergens et al., 2024; Liang et al., 2019; Chen et al., 2023b; Qu et al., 2024; Yoon & Kim, 2024). However, a critical gap remains: EDL is fundamentally a deep learning model, and the

---

\*Corresponding author.
[1]Code is available at https://github.com/LinyeLi60/M-EDL

behavior of deep learning models is strongly shaped by optimization processes such as loss function design and gradient dynamics. Thus, treating EDL merely as a universally applicable probabilistic estimator, while overlooking its optimization characteristics, provides an incomplete understanding.

In this paper, we revisit EDL from an optimization perspective and uncover a non-trivial theoretical connection: minimizing the expected cross-entropy loss under the Dirichlet prior implicitly drives the solution towards that of a multi-class Support Vector Machine (SVM) (Crammer & Singer, 2001; Hsu & Lin, 2002), which maximizes the decision margin. This observation reveals that the notion of *margin* naturally emerges from the EDL objective, providing a more fundamental and optimization-aligned interpretation of uncertainty than conventional metrics derived solely from Dirichlet parameters. Intuitively, samples with large margins are more certain, while those near the decision boundary are intrinsically uncertain. Building on this insight, we develop a principled framework that bridges optimization and uncertainty estimation in EDL. Our contributions are summarized as:

- **Theoretical connection to maximum-margin SVMs.** We formally establish a connection between Evidential Deep Learning (EDL) and the maximum-margin principle of Support Vector Machines (SVMs). In particular, we prove that the Uncertainty-aware Cross-Entropy (UCE) loss admits a margin-aware lower bound, offering a new interpretation of EDL beyond its probabilistic formulation.

- **Optimization-consistency principle for uncertainty.** We introduce the notion of *optimization-consistency*, which defines an uncertainty measure as valid if its value monotonically decreases as a sample approaches the global optimum of the training objective. This principle bridges loss optimization with predictive uncertainty, providing a new criterion for evaluating uncertainty measures.

- **Margin-aware predictive uncertainty (MPU).** Guided by the optimization-consistency principle, we propose a novel uncertainty measure, *Margin-aware Predictive Uncertainty (MPU)*, which is explicitly aligned with the UCE loss and captures the margin between the dominant class evidence and the aggregated evidence of other classes.

- **Experimental validation.** Extensive experiments on OOD detection and misclassification detection benchmarks demonstrate that aligning uncertainty estimation with optimization objectives substantially improves predictive reliability.

## 2 RELATED WORKS

### 2.1 SUBJECTIVE LOGIC AND EVIDENTIAL DEEP LEARNING

The theoretical foundation of Evidential Deep Learning (EDL) is closely related to Dempster-Shafer Theory (DST) (Dempster, 1968) and Subjective Logic (Jøsang, 2016), which extend conventional probability theory by explicitly modeling uncertainty. A central concept in this framework is the *subjective opinion* $\boldsymbol{\omega} = (\boldsymbol{b}, u, \boldsymbol{a})$, where $\boldsymbol{b} = (b_1, \ldots, b_K)$ denotes the *belief* masses, $u \in [0, 1]$ represents the uncertainty mass, and $\boldsymbol{a} = (a_1, \ldots, a_K)$ specifies the prior base-rate distribution. These quantities satisfy $\sum_{j=1}^{K} b_j + u = 1$ and $\sum_{j=1}^{K} a_j = 1$. Intuitively, $\boldsymbol{b}$ reflects direct evidence for each class, $u$ quantifies the lack of evidence, and $\boldsymbol{a}$ serves as an uninformative prior. Building upon this framework, for input $\boldsymbol{x}$, EDL (Sensoy et al., 2018) maps the network parameterized with $\boldsymbol{\Theta}$ to logits $\boldsymbol{z}$, followed by a non-negative activation function $\sigma$, to get the evidence value, i.e., $\boldsymbol{e} = \sigma(\boldsymbol{z}(\boldsymbol{x}; \boldsymbol{\Theta}))$. Then, the corresponding Dirichlet parameters are defined as $\boldsymbol{\alpha} = \boldsymbol{e} + 1$, with total strength $S = \sum_{k=1}^{K} \alpha_k$. From these parameters, the *belief* and uncertainty masses can be computed as $b_j = e_j/S$ and $u = K/S$. To train the model, EDL minimizes the expected empirical risk under the Dirichlet distribution, using losses such as the Brier Score, Type II Maximum Likelihood loss, and the Expected Cross-entropy loss. A detailed derivation is provided in Appendix A. However, prior work has typically optimized EDL models using a single loss function without formally establishing why that loss is particularly suitable. In this work, we focus on the expected cross-entropy over the Dirichlet distribution, commonly referred to as the *Uncertainty-aware Cross-Entropy (UCE)* loss (Biloš et al., 2019; Sensoy et al., 2018; Charpentier et al., 2020; 2021):

$$\mathcal{L}_{\text{UCE}}(\boldsymbol{\Theta}) = \int \left[ \sum_{j=1}^{K} -y_{ij} \log(p_{ij}) \right] \frac{1}{B(\boldsymbol{\alpha}_i)} \prod_{j=1}^{K} p_{ij}^{\alpha_{ij}-1} d\boldsymbol{p}_i = \sum_{j=1}^{K} y_{ij}(\psi(S_i) - \psi(\alpha_{ij})), \quad (1)$$

where $\psi(\cdot)$ denotes the digamma function and $B(\cdot)$ is the multivariate Beta function. We will show that optimization under this loss exhibits properties: the learned model parameters naturally align with the maximum-margin solutions of classical multi-class Support Vector Machines (SVMs). This reveals a principled link between EDL and the well-established theory of max-margin learning.

## 2.2 UNCERTAINTY MEASURES WITH DIRICHLET DISTRIBUTIONS

In EDL, uncertainty is quantified by exploiting the probabilistic properties of the Dirichlet distribution. Several uncertainty measures have been proposed in previous works, each capturing different aspects of predictive reliability. First, the **Vacuity of Evidence (VoE)**, defined as $u = K/S$ with $S = \sum_{j=1}^{K} \alpha_j$, originates from Subjective Logic (Jøsang, 2016). It measures the overall lack of evidence in favor of any class, directly reflecting the ignorance of the model with respect to the prediction. Second, the **Differential Entropy (DE)** of the Dirichlet distribution, $\mathrm{ENT}(\mathrm{Dir}\,(\boldsymbol{p} \mid \boldsymbol{\alpha}))$, is adopted from information theory as a global measure of the dispersion of the predictive distribution, with higher entropy indicating greater uncertainty. Third, **Mutual Information (MI)**, $\mathbb{I}(y, \boldsymbol{p})$, is adapted from Bayesian Model Averaging (Malinin & Gales, 2018; Gal et al., 2016) to isolate epistemic uncertainty from total predictive uncertainty, quantifying how much knowledge of the model parameters reduces uncertainty in predictions. A detailed derivation of these measures is provided in Appendix B. Despite the availability of various measures, prior work has not established which is most effective. In this paper, we analyze and compare them from an optimization perspective for EDL models trained with the UCE loss.

## 3 PRELIMINARY

### 3.1 NOTATIONS

Following Juergens et al. (2024), we decompose the overall parameter set of the model, $\boldsymbol{\Theta}$, into the last-layer linear classifier $\boldsymbol{W} := \{\boldsymbol{w}_1, \ldots, \boldsymbol{w}_K\} \in \mathbb{R}^{K \times d}$ and the parameters of the feature embedding function $\boldsymbol{\Psi}$ that maps input $\boldsymbol{x}$ into $d$-dimensional latent feature. The evidence for each class is defined as $e_k(\boldsymbol{x}; \boldsymbol{\Theta}) = \exp(z_k) = \exp\{\boldsymbol{w}_k^\top \boldsymbol{\Psi}(\boldsymbol{x})\}$, where $\boldsymbol{w}_k$ denotes the classifier vector associated with class $k$. The exponential function $\exp(\cdot)$ serves as a non-negative activation, ensuring that the predicted evidence remains positive.

### 3.2 CRAMMER AND SINGER $K$-CLASS SVM

In the following, we will analyze the relationship between EDL and a classical multi-class support vector machine (SVM), namely the Crammer and Singer (C&S) $K$-class SVM. To this end, we first recall the primal formulation of this SVM and present the form of its optimal solution.

**Definition 1.** The primal optimization problem for the Crammer and Singer multi-class SVM (Crammer & Singer, 2001) is defined as

$$\min_{\boldsymbol{w}, \boldsymbol{\xi}} \quad \frac{\beta}{2} \sum_{j=1}^{K} \boldsymbol{w}_j^\top \boldsymbol{w}_j + \sum_{i=1}^{N} \xi_i \tag{2}$$
$$\text{s.t.} \quad \boldsymbol{w}_{y_i}^\top \boldsymbol{x}_i - \boldsymbol{w}_j^\top \boldsymbol{x}_i + \delta_{y_i, j} \geq 1 - \xi_i, \quad \forall i \in [1, N],\ j \neq y_i,$$

where $\beta > 0$ is the regularization constant, $N$ is the number of training samples, $\xi_i \geq 0$ are slack variables, $y_i \in [K]$ denotes the ground-truth label of sample $\boldsymbol{x}_i$, $\delta$ is the Kronecker delta function $\delta_{p,q} = 1$ if $p = q$ else 0 otherwise. A sample lies correctly outside the margin if $\xi_i = 0$, while $\xi_i > 0$ indicates either a margin violation or a misclassified sample. By considering the corresponding dual problem and applying the Karush–Kuhn–Tucker (KKT) conditions (Bertsekas, 1997), the optimal classifier vectors can be expressed as a weighted sum of the training samples

$$\boldsymbol{w}_j = \beta^{-1} \sum_{i=1}^{N} \left(\delta_{y_i, j} - \eta_{i,j}\right) \boldsymbol{x}_i, \tag{3}$$

where $\eta_{ij}$ are the dual coefficients satisfying $\eta_{i,1}, \ldots, \eta_{i,K} \geq 0$ and $\sum_{j=1}^{K} \eta_{i,j} = 1$. A complete derivation of the dual formulation and Eq. 3 is provided in Appendix C.1. The term $(\delta_{y_i, j} - \eta_{i,j})$

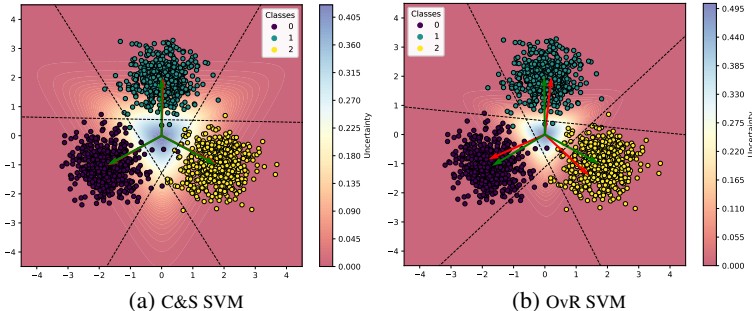

(a) C&S SVM        (b) OvR SVM

Figure 1: Visualization of predictive uncertainty (Vacuity of Evidence), optimized linear classifiers $\boldsymbol{W}$ of EDL and C&S SVMs on a toy dataset in (a). We also trained a One-vs-Rest (OvR) SVM (Schölkopf, 1997) on this dataset for comparison in (b).

acts as a sample contribution weight. Let's analyze this contribution: For the correct class ($j = y_i$), the weight is $(1 - \eta_{i,y_i})$. If sample $\boldsymbol{x}_i$ is easily classified (i.e., not a support vector), $\eta_{i,y_i} \to 1$, and its contribution $\boldsymbol{w}_{y_i}$ vanishes. This means *easy samples do not define the classifier*. For any incorrect class ($j \neq y_i$), the weight is $(-\eta_{i,j})$. If $\eta_{i,j} > 0$, it implies $\boldsymbol{x}_i$ is a *support vector* that is on or within the margin of class $j$. The term $-\eta_{i,j}\boldsymbol{x}_i$ then acts to "push" the classifier $\boldsymbol{w}_j$ away from this incorrectly active sample. In essence, the optimal SVM classifier $\boldsymbol{w}_j$ is constructed by "pulling" towards challenging samples of its own class and "pushing" away from samples of other classes that violate the margin. The magnitude of these pulls and pushes is determined by the dual variables $\eta_{ij}$, which are non-zero only for the *support vectors*, where the most informative or *uncertain* samples for defining the decision boundary.

## 4    MARGIN-AWARE PROPERTY OF EDL OPTIMIZED WITH UCE LOSS

In this section, we will show that the classifiers $\boldsymbol{W}$ obtained by minimizing the UCE loss takes a form closely resembling the weight vector of a $K$-class SVM. This observation suggests that EDL implicitly performs margin-based classification, thus providing a connection between probability-based uncertainty estimation and the classical principle of maximum margin.

### 4.1    MOTIVATION: ALIGNMENT OF EDL CLASSIFIERS WITH SVM OPTIMAL SOLUTIONS

To analyze the relationship between C&S SVM classifiers and those learned by an EDL model (via Eq. 1), we conducted a simple experiment. We assumed fixed, Gaussian-distributed features—an approach that enables direct comparison with SVMs. Importantly, these results are largely independent of the specific distribution, as the classifiers are mainly determined by support vectors. Further visualizations with different distributions are available in Appendix E.

In Fig. 1, we conduct experiments on a toy dataset consisting of three Gaussian-distributed clusters with an isotropic standard deviation of 0.3. We assume that the EDL model's mapping $\boldsymbol{\Psi}$ is the identity function. This allows us to focus solely on the properties of the loss function with respect to the classifier weight vectors $\boldsymbol{W}$. Green arrows represent the classifiers learned by the EDL model, while red arrows indicate the directions of the SVM classifier vectors. Black dotted lines correspond to the separating hyperplanes learned by the SVMs. We can make three main conclusions:

1. In Fig. 1a, the EDL classifiers closely align with the C&S SVM classifiers, suggesting an implicit margin-aware separation in the EDL optimization with UCE loss.

2. In contrast, when comparing EDL with the classical one-vs-rest (OvR) SVM (Schölkopf, 1997) as shown in Fig. 1b, the alignment is less tight. While EDL and OvR classifiers share similar overall behavior in this problem, the final OvR solution $W$ is not as closely aligned with the EDL classifier as C&S SVM is, highlighting that the tight alignment is specific to the C&S multi-class formulation.

3. By visualizing the uncertainty adopted in EDL (Sensoy et al., 2018) (i.e., the vacuity of evidence $u = K/S$), we find that regions of high uncertainty correspond to the intersection of the negative half-spaces defined by the decision hyperplanes, as shown in Fig. 1a.

These observations are heuristic, motivating us to explore a potential implicit connection between EDL and the C&S SVM, with the aim of gaining a deeper understanding of the underlying mechanism of EDL. In the following, we analyze the relationship between the optimization objectives of EDL and those of the C&S SVM, as well as their respective optimal solutions.

## 4.2 The Margin-aware Property of Evidential Deep Learning Loss

Based on our previous observations, we first demonstrate that the UCE loss in Eq. 1 satisfies a margin-aware property by explicitly optimizing a margin between the logits for the ground-truth class and those of the incorrect classes.

**Theorem 1. The Margin-Aware Property of UCE Loss.** For an input sample $x$ with a corresponding ground-truth label $y$, the UCE loss $\mathcal{L}_{\mathrm{UCE}}$ can be lower-bounded in the following form:

$$\mathcal{L}_{\mathrm{UCE}}\left(x, y, W, \Psi\right) \geq \phi\left(\sum_{j \neq y}\left(w_j^\top \Psi(x) - w_y^\top \Psi(x)\right)\right) = \phi\left(-\mathcal{M}(x, y; W, \Psi)\right), \quad (4)$$

where $\phi(t) = \log\left(1 + (K-1)\min\left(1, \exp\left(t\right)\right)\right)$. Let $z_j = w_j^\top \Psi(x)$ for simplicity. The inequality achieves equality when $z_j = z_{j'}$ for all $j, j' \neq y$, and $\mathcal{M}\left(x, y; W, \Psi\right) = \sum_{j \neq y}\left(w_y^\top \Psi(x) - w_j^\top \Psi(x)\right)$. The full derivation of the margin-aware lower bound is provided in Appendix C.2. Since $\phi(t)$ is an increasing function with respect to $t$, minimizing the UCE loss is equivalent to maximizing the margin-like quantity $(K-1)z_y - \sum_{j \neq y} z_j$, which contrasts the ground-truth class output $z_y$ simultaneously to all the other outputs $z_j$ for all $j \neq y$. Thus, we call Eq. 4 a *margin-aware property*. Maximizing $\mathcal{M}(x, y)$ encourages $z_y$ to be positive and relatively large, while driving each $z_j$ ($j \neq y$) to be negative and relatively small.

**Proposition 1. On the Analogy between EDL Gradient Dynamics and the SVM Dual Solution.** The optimization dynamics of Evidential Deep Learning (EDL) classifiers reveal a striking parallel to the structure of the C&S SVM's optimal dual solution (Eq. 3). This suggests that EDL's learning process inherently mimics the margin-maximizing mechanism of SVMs. To demonstrate this, we analyze the gradient of the UCE loss with respect to a classifier weight vector $w_j$. Assuming the Dirichlet strength for the correct class is large ($\alpha_{i,y_i} \gg 1$), a condition typically met by using an exponential activation, the gradient contribution from each sample can be approximated as:

$$\nabla_{w_j} \mathcal{L}_{\mathrm{UCE}}(W, \Psi) \approx \sum_{i=1}^N \left(\delta_{y_i, j} - b_{ij}\right) \Psi(x_i), \quad \text{where } b_{ij} = \frac{\alpha_{ij} - 1}{S_i}. \quad (5)$$

A complete derivation process is in Appendix C.3. Here, $b_{ij}$ is the belief mass assigned to class $j$ for sample $i$, and $\Psi(x_i)$ is the sample's feature embedding. The analogy becomes clear when we compare the core components:

- SVM Optimal Solution (Eq. 3): $w_j \propto \sum_i (\delta_{y_i, j} - \eta_{i,j}) x_i$
- UCE Loss Gradient Update (Eq. 5): $\Delta w_j \propto -\nabla_{w_j} \mathcal{L} \approx \sum_i (\delta_{y_i, j} - b_{i,j}) \Psi(x_i)$

The belief mass $b_{ij}$ in the EDL gradient plays a role directly analogous to the dual coefficient $\eta_{ij}$ in the SVM solution. Both quantify a sample's relationship to each class. Let's examine the update term $(\delta_{y_i, j} - b_{i,j})$: For the correct class ($j = y_i$), the term is $(1 - b_{i,y_i})$, where lower belief $b_{i,y_i}$ means higher uncertainty about the ground truth label. Since the gradient update is small for confident samples ($b_{i,y_i} \to 1$), the learning process naturally focuses on samples where the model remains uncertain. For any incorrect class ($j \neq y_i$), the term is $(-b_{i,j})$. This is the negative belief in that wrong class. If the model incorrectly assigns high belief to class $j$ ($b_{ij} > 0$), this term generates a strong gradient to "push" $w_j$ away from the sample's embedding $\Psi(x_i)$. This mechanism is a dynamic equivalent to the SVM's optimal solution, where C&S SVM identifies a fixed set of support vectors ($\eta_{ij} > 0$) to define the boundary, and EDL's gradient descent continuously uses the belief masses ($b_{ij}$) to achieve a similar outcome. Samples with high uncertainty (low $b_{i,y_i}$) or high

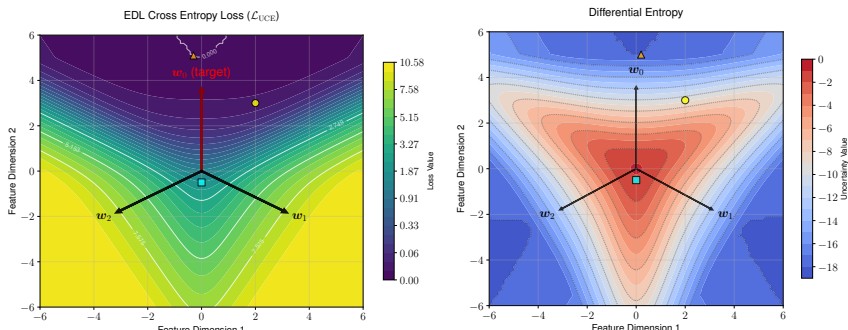

Figure 2: The desired alignment between the loss landscape and predictive uncertainty. The figure compares the UCE Loss landscape (left) with the predictive differential entropy (right). As a sample moves from an ambiguous region (cyan square at the origin) towards an ideal region for the target class $w_0$ (orange triangle), both its loss and uncertainty should systematically decrease.

conflict (high $b_{i,j}$ for $j \neq y_i$) act as "dynamic support vectors", exerting the strongest influence on the classifier updates. This demonstrates that EDL's optimization process is inherently structured to promote large margin separation, akin to a multi-class C&S SVM.

### 4.3 OPTIMIZATION-CONSISTENCY PROPERTY OF UNCERTAINTY MEASURE

In the previous subsection, we established that minimizing the UCE loss in EDL implicitly aligns the learned classifier with the maximum-margin solution of SVMs. This raises a natural question: *which uncertainty measures are compatible with such an optimization process?* While many uncertainty measures have been proposed in prior works (e.g., vacuity of evidence, differential entropy and mutual information), not all of them necessarily reflect the progress of the optimization dynamics. Intuitively, an appropriate uncertainty measure should achieve its minimum value on a sample exactly when this sample aligns with the optimal solution of the underlying loss function curve. Otherwise, the measure may contradict the training objective and thus misrepresent predictive reliability. To formalize this requirement, we introduce the principle of *Optimization-Consistency*, which characterizes uncertainty measures whose values monotonically decrease as the optimization drives a sample toward the global optimum of the UCE loss.

**Theorem 2. Optimization-Consistency Property of Uncertainty.** Let $u(\boldsymbol{x}; \boldsymbol{W}, \boldsymbol{\Psi})$ be an uncertainty measure induced by the network parameters set $\{\boldsymbol{W}, \boldsymbol{\Psi}\}$. We say that $u$ is *optimization-consistent* with respect to the UCE loss if, for any pair of training sample $(\boldsymbol{x}, y)$ and $(\boldsymbol{x}', y')$, the following implication holds:

$$\mathcal{L}_{\text{UCE}}(\boldsymbol{x}, y, \boldsymbol{W}, \boldsymbol{\Psi}) \leq \mathcal{L}_{\text{UCE}}(\boldsymbol{x}', y', \boldsymbol{W}, \boldsymbol{\Psi}) \implies u(\boldsymbol{x}; \boldsymbol{W}, \boldsymbol{\Psi}) \leq u(\boldsymbol{x}'; \boldsymbol{W}, \boldsymbol{\Psi}). \quad (6)$$

In other words, as the optimization dynamics drive the sample closer to the global optimum of the objective loss, its predictive uncertainty should monotonically decrease. One can envision the loss function as a surface, where individual samples map to distinct points on this surface. A meaningful uncertainty measure should yield lower values as samples are closer to the *valley*, i.e., the optimal point of the loss surface.

Let's further explore whether existing EDL-related loss functions are characterized by this Optimization-Consistency property. Intuitively, we discover that the Vacuity of Evidence, defined as $u(\boldsymbol{x}; \boldsymbol{W}, \boldsymbol{\Psi}) = K / \sum_{j=1}^{K} \exp(\boldsymbol{w}_j^\top \boldsymbol{\Psi}(\boldsymbol{x}) + 1)$, does not fulfill this property. A counter-example demonstrates this clearly. Suppose we have $K = 3$ classes. For samples $\boldsymbol{x}$ and $\boldsymbol{x}'$, both labeled as $y = y' = 1$, assuming their predicted Dirichlet distributions are $\boldsymbol{\alpha} = (10, 1, 1)$ and $\boldsymbol{\alpha}' = (10, 10, 1)$, respectively. It is clear that the first sample has a lower loss value. However, according to the definition of Vacuity of Evidence, the second sample $\boldsymbol{x}'$ appears to have lower uncertainty as $3/(10+1+1) > 3/(10+10+1)$. Hence, we conclude that the Vacuity of Evidence fails to meet the Optimization-Consistency property. We further discover that the Differential Entropy, defined as in Eq. 7, satisfies the above-mentioned property.

**Proposition 2. Optimization-Consistency of Differential Entropy.** Let $\mathrm{ENT}(\mathrm{Dir}(\boldsymbol{p} \mid \boldsymbol{\alpha}))$ denote the differential entropy of the Dirichlet distribution parameterized by $\boldsymbol{\alpha}$ as

$$\mathrm{ENT}(\mathrm{Dir}(\boldsymbol{p} \mid \boldsymbol{\alpha})) = -\int_{\boldsymbol{p} \in \Delta^{K-1}} \mathrm{Dir}(\boldsymbol{p} \mid \boldsymbol{\alpha}) \log \mathrm{Dir}(\boldsymbol{p} \mid \boldsymbol{\alpha}) \, d\boldsymbol{p}$$

$$= \log B(\boldsymbol{\alpha}) + (S - K)\psi(S) - \sum_{j=1}^{K} (\alpha_j - 1)\psi(\alpha_j). \tag{7}$$

Then the differential entropy uncertainty measure is optimization-consistent with the UCE loss (Theorem 2). That is, for any two training samples $(\boldsymbol{x}, y)$ and $(\boldsymbol{x}', y')$, we have

$$\mathcal{L}_{\mathrm{UCE}}(\boldsymbol{x}, y, \boldsymbol{W}, \boldsymbol{\Psi}) \leq \mathcal{L}_{\mathrm{UCE}}(\boldsymbol{x}', y', \boldsymbol{W}, \boldsymbol{\Psi}) \implies \mathrm{ENT}(\mathrm{Dir}(\boldsymbol{p} \mid \boldsymbol{\alpha}(\boldsymbol{x}))) \leq \mathrm{ENT}(\mathrm{Dir}(\boldsymbol{p} \mid \boldsymbol{\alpha}(\boldsymbol{x}'))). \tag{8}$$

Complete proof can be in Appendix C.4. It should be noted that the above equation is one-sided only: while reducing the UCE loss reliably decreases entropy, entropy by itself is insufficient to determine the relative ordering of UCE loss across samples. To provide a more intuitive perspective on this concept, we conduct an experiment on a three-class toy dataset. As demonstrated in Fig. 2, we chart the loss contour of UCE and the corresponding differential entropy contour. Notably, points located closer to the global optimum location of the loss function (depicted by the orange triangle) exhibit a lesser uncertainty value compared to those positioned farther away from the global optimum (indicated by the cyan square). This observation underscores our assertion that the differential entropy measure displays optimization consistency with respect to the UCE loss.

## 4.4 MARGIN-AWARE PREDICTIVE UNCERTAINTY

Inspired by the margin-aware property of UCE loss and the Optimization-Consistency Principle, we propose a specially designed uncertainty specifically for UCE loss, which we refer to as Margin-Aware Predictive Uncertainty (MPU).

**Proposition 3. The Margin-Aware Predictive Uncertainty (MPU)** is an uncertainty measure inspired by the optimization-consistency principle and the margin-aware property of the UCE loss. We define the MPU by the margin of the Dirichlet parameters $\boldsymbol{\alpha}$ as

$$\mathrm{MPU}(\boldsymbol{\alpha}) = (K - 1)\alpha_{\hat{y}} - \sum_{j \neq \hat{y}} \alpha_j, \tag{9}$$

where $\hat{y}$ denotes the predicted class label corresponding to the highest probability in the model's output. Intuitively, this formulation measures the margin between the evidence for the predicted class and all other classes. Therefore, a higher MPU score signifies greater predictive certainty.

**Characteristics and Advantages of MPU.** To illustrate the characteristics and advantages of MPU over conventional uncertainty metrics, we provide a comparative analysis in Fig. 3. From left to right, the predictive distribution becomes more concentrated on the ground-truth class. According to the optimization consistency principle, the measured uncertainty should monotonically decrease. However, we observe the following behaviors from the conventional metrics: (1) VoE and MI remain stagnant, failing to reflect the increasing certainty. (2) DE shows a moderate decrease, but its negative value is less intuitive. (3) MPU increases significantly from 4 to 28, providing a sensitive and interpretable measure of confidence. Therefore, the MPU possesses several distinct advantages:

- **Intuitive and Interpretable**: MPU is more interpretable than Differential Entropy (DE). DE values are non-positive and lack an intuitive scale, making the practical difference between values like -1.9 and -2.9 unclear. In contrast, MPU starts at 0 for maximum uncertainty and increases with predictive confidence, offering a direct and monotonic measure.

- **Sensitivity**: Unlike MI, which remains stagnant, or DE, which shows only moderate variation, MPU is highly sensitive to the concentration of a probability distribution. This is evidenced by its dramatic increase from 4 to 28 as model confidence grows, proving its superior capacity for discriminating between confidence levels.

- **Versatility**: VoE is limited to OOD detection, as it only captures uncertainty from evidence absence while ignoring the evidential distribution for a fixed total Dirichlet strength ($S$). In contrast, MPU is more versatile. It measures uncertainty from both insufficient evidence (for OOD detection) and conflicting class evidence (e.g., when top predictions are close), making it effective for misclassification detection as well.

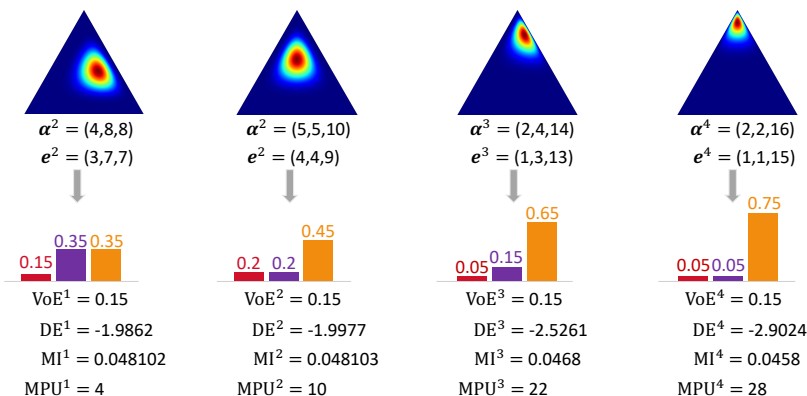

Figure 3: Illustration of MPU's advantages over conventional uncertainty metrics Vacuity of Evidence (VoE), Differential Entropy (DE) and Mutual Information (MI).

## 5 Experiments

In this section, we empirically validate our theoretical findings regarding the role of optimization-consistency principle in uncertainty estimation. Specifically, we perform experiments on CIFAR-10 and CIFAR-100, with out-of-distribution (OOD) detection and misclassification detection tasks.

Table 1: Summary of EDL-related methods, highlighting their loss functions, activation functions, post-hoc regularization on the predictive Dirichlet distribution, and additional training techniques.

| Method | Loss Function | Activation Function $\sigma$ | Post-Hoc Regularization | Additional Techniques |
|---|---|---|---|---|
| EDL | Brier Score | Softplus | KL Divergence | - |
| RED | T2ML | Exp | KL Divergence | Evidence Regularization |
| $\mathcal{I}$-EDL | Brier Score | Softplus | KL Divergence | Fisher Information Regularization |
| R-EDL | Brier Score | Softplus | KL Divergence | Adjust Prior |
| Re-EDL | Brier Score | Softplus | - | Adjust Prior |
| PostN | UCE | Exp | Entropy Regularization | Multiple Normalization Flow |
| NatPN | UCE | Exp | Entropy Regularization | Normalization Flow |
| Our | UCE | Exp | - | - |

### 5.1 Experimental Implementation

**Baseline Methods.** We benchmark our approach against two sets of baselines: 1) Evidential methods, including EDL, PostN, NatPN, and recent EDL variants (RED, $\mathcal{I}$-EDL, R-EDL, Re-EDL); and 2) Classical uncertainty methods, like MC Dropout and DUQ. Following their original designs, we use Vacuity of Evidence (VoE, defined as $K/S$) as the uncertainty measure for EDL and its variants, and Dirichlet strength ($S$) for PostN and NatPN. Table 1 provides a summary of these baselines.

**Implementation Details.** We conduct experiments on the CIFAR-10/100 datasets, using SVHN (Netzer et al., 2011), GTSRB (Stallkamp et al., 2012), Places365 (Zhou et al., 2017), and Food101 (Bossard et al., 2014) as OOD datasets. Following Chen et al. (2025), we employ VGG16 as the backbone network. Models are trained for 200 epochs using SGD optimizer with an initial learning rate of 0.1, decayed by a cosine scheduler. Notably, our training objective relies solely on the UCE loss and does not involve any post-hoc regularization terms, such as KL regularization, to constrain the predicted Dirichlet parameters or suppress misleading evidence. Performance on OOD and misclassification detection is evaluated using the area under the precision-recall curv (AUPR).

### 5.2 Experimental Results Analysis

**MPU as the Best Companion for UCE Loss.** In Table 2, we compare UCE loss combined with four different uncertainty measures: MPU (Margin-aware Predictive Uncertainty), DE (Differential

Table 2: Performance comparison evaluated by classification accuracy and AUPR scores for OOD detection and misclassification detection on CIFAR-10. $\rightarrow$ X indicates using X as OOD data. Results are averaged over 5 runs, with mean and standard deviation reported. The bold numbers indicate the best performance, while the underlined numbers represent the second-best method.

| Method | →SVHN | →CIFAR100 | →GTSRB | →Places365 | →Food101 | Cls Acc | Mis Detect |
|---|---|---|---|---|---|---|---|
| MC Dropout | 78.40±3.88 | 85.39±0.58 | 83.51±2.32 | 67.63±2.34 | 76.07±2.58 | 90.16±0.23 | 98.86±0.06 |
| EDL | 82.32±1.21 | 87.13±0.26 | 84.57±1.26 | 70.46±0.77 | 80.18±0.69 | 88.48±0.32 | 98.74±0.07 |
| DUQ | 81.44±4.63 | 85.38±0.37 | 83.35±4.30 | 66.20±3.61 | 75.87±4.40 | 89.39±0.13 | 97.98±0.34 |
| PostN | 83.76±0.46 | 87.07±0.94 | 84.83±1.67 | 71.79±2.36 | 78.83±2.53 | 87.82±0.06 | 97.46±0.06 |
| NatPN | 83.56±0.38 | 88.05±0.75 | 84.76±2.03 | 71.14±2.06 | 79.44±2.63 | 87.73±0.09 | 97.53±0.05 |
| RED | 82.85±2.35 | 87.84±0.54 | 85.30±3.31 | 70.78±2.33 | 79.91±1.92 | 89.43±0.28 | 98.82±0.09 |
| $\mathcal{I}$-EDL | 84.97±2.11 | 86.31±0.32 | 84.79±2.13 | 68.92±1.21 | 77.75±0.27 | 88.38±0.15 | 98.71±0.11 |
| R-EDL | 85.00±1.22 | 87.73±0.31 | 87.25±0.69 | 71.97±0.69 | 79.64±2.36 | 90.09±0.31 | 98.98±0.05 |
| Re-EDL | **89.94±1.40** | 88.31±0.16 | **90.53±2.04** | **73.42±1.05** | **80.83±1.72** | 90.13±0.25 | 98.81±0.05 |
| Our /w VoE | 48.96±8.96 | 66.45±1.73 | 69.64±2.18 | 43.21±3.00 | 45.65±2.26 | | 96.63±0.32 |
| Our /w MI | 84.28±2.60 | 86.73±0.40 | 86.35±2.53 | 68.61±1.93 | 76.81±1.58 | | 99.09±0.08 |
| Our /w DE | 87.32±1.20 | 88.11±0.34 | 87.30±2.40 | 70.91±1.58 | 78.64±1.40 | **93.35±0.25** | 99.31±0.06 |
| Our /w MPU | 87.36±0.91 | **88.92±0.31** | 88.71±1.85 | 72.82±1.24 | 79.79±1.35 | | **99.41±0.04** |
| Δ (MPU vs VoE) | + 38.40 | +22.47 | +19.06 | +29.69 | +34.14 | | +2.78 |

Table 3: Comparison of uncertainty measures for OOD and misclassification detection on CIFAR-100, evaluated by AUPR and classification accuracy. PostN[†] indicates that the pre-trained feature encoder is frozen and only the density estimator is trained follow Kirichenko et al. (2020).

| Method | CIFAR100→SVHN | | | | CIFAR100→GTSRB | | | |
|---|---|---|---|---|---|---|---|---|
| | MPU | DE | MI | VoE | MPU | DE | MI | VoE |
| EDL | 65.02±7.82 | 65.63±7.92 | 65.70±7.93 | 65.71±7.93 | 69.42±4.41 | 69.58±4.62 | 69.60±4.64 | 69.60±4.64 |
| $\mathcal{I}$-EDL | 40.72±15.22 | 45.10±18.72 | 42.71±16.18 | 42.26±15.72 | 58.42±15.26 | 53.00±11.76 | 51.50±10.72 | 51.29±10.57 |
| R-EDL | 64.91±5.76 | 65.35±5.86 | 65.69±5.90 | 65.74±5.86 | 73.61±5.59 | 73.83±5.58 | 73.98±5.54 | 73.98±5.51 |
| Re-EDL | 65.20±6.03 | 65.97±6.04 | 66.08±6.03 | **66.09±6.03** | 72.97±4.20 | 73.14±4.41 | 73.15±4.43 | 73.15±4.44 |
| PostN | 58.62±6.03 | 58.62±5.94 | 26.84±4.13 | 26.84±5.41 | 54.49±8.13 | 56.28±7.65 | 36.58±4.69 | 53.81±6.88 |
| PostN[†] | 45.84±2.58 | 49.25±1.41 | 25.72±1.44 | 45.12±1.54 | 71.36±1.77 | 69.69±1.74 | 50.71±1.61 | 71.65±1.86 |
| Our | 60.46±6.85 | 57.76±8.32 | 54.41±8.48 | 49.74±6.53 | **76.06±1.69** | 71.73±4.76 | 61.92±4.91 | 56.05±3.38 |
| Method | CIFAR100→Places365 | | | | CIFAR100→Food101 | | | |
| | MPU | DE | MI | VoE | MPU | DE | MI | VoE |
| EDL | 45.45±1.47 | 45.52±1.52 | 45.52±1.52 | 45.52±1.52 | 68.54±2.19 | 68.87±2.46 | 68.89±2.51 | 68.89±2.51 |
| $\mathcal{I}$-EDL | 31.16±10.22 | 28.34±7.25 | 26.77±5.62 | 26.53±5.37 | 46.95±19.68 | 42.06±14.68 | 38.90±11.42 | 38.40±10.9 |
| R-EDL | 45.66±1.43 | 45.70±1.44 | 45.71±1.46 | 45.71±1.46 | 70.15±0.83 | 70.54±0.82 | 70.76±0.78 | **70.77±0.76** |
| Re-EDL | 43.88±2.10 | 43.92±2.13 | 43.91±2.14 | 43.91±2.14 | 69.99±1.30 | 70.46±1.46 | 70.50±1.48 | 70.50±1.48 |
| PostN | 31.88±3.40 | 33.04±4.54 | 18.05±3.12 | 31.64±3.79 | 47.55±5.98 | 50.19±7.84 | 22.79±4.45 | 46.85±5.82 |
| PostN[†] | 38.41±1.23 | 39.52±1.29 | 19.96±1.14 | 38.01±1.54 | 56.33±1.90 | 56.87±1.81 | 26.09±1.00 | 55.72±1.73 |
| Our | **49.62±1.05** | 45.01±1.40 | 33.27±0.91 | 27.83±0.37 | 68.34±0.89 | 66.75±2.03 | 55.40±2.18 | 44.72±1.27 |
| Method | CIFAR100→CIFAR10 | | | | CIFAR-100 Mis Detect | | | |
| | MPU | DE | MI | VoE | MPU | DE | MI | VoE |
| EDL | 72.75±1.02 | 72.70±1.10 | 72.67±1.12 | 72.67±1.12 | 91.05±0.29 | 90.56±0.32 | 90.48±0.33 | 90.47±0.33 |
| $\mathcal{I}$-EDL | 60.29±11.01 | 57.01±8.08 | 55.54±6.58 | 55.29±6.33 | 60.29±11.01 | 57.01±8.08 | 55.54±6.58 | 55.29±6.33 |
| R-EDL | 73.16±0.78 | 73.11±0.78 | 73.00±0.80 | 72.95±0.81 | 90.58±0.29 | 90.18±0.32 | 89.90±0.33 | 89.83±0.33 |
| Re-EDL | 71.69±1.49 | 71.55±1.63 | 71.52±1.65 | 71.51±1.65 | 90.92±0.26 | 90.36±0.29 | 90.26±0.29 | 90.26±0.29 |
| PostN | 59.89±5.42 | 62.15±7.55 | 41.76±6.56 | 59.06±7.92 | 79.75±5.34 | 81.06±5.12 | 44.16±4.56 | 78.45±1.53 |
| PostN[†] | 68.07±1.87 | 68.27±1.45 | 44.87±1.65 | 67.40±1.42 | 88.57±1.58 | 89.27±1.53 | 61.94±1.25 | 87.87±1.15 |
| Our | **76.05±0.49** | 74.55±0.48 | 65.77±0.88 | 58.72±1.35 | **94.34±0.25** | 90.34±0.90 | 80.71±1.11 | 76.41±1.13 |

Entropy), MI (Mutual Information), and VoE (Vacuity of Evidence) on CIFAR-10 OOD detection and misclassification detection tasks. Several key observations emerge from this comparison.

(1) The combination of VoE with UCE loss and an exponential activation performs the worst overall, as it violates the previously discussed optimization-consistency principle. This finding corroborates the known overconfidence issue of UCE reported in EDL, which explains why subsequent EDL related studies have avoided this combination.

(2) MPU achieves the best performance when combined with UCE loss, consistently outperforming the other uncertainty measures. Among the alternatives, Differential Entropy, which also satisfies optimization consistency, provides the second-best results. This supports our claim that the effectiveness of an uncertainty measure critically depends on its consistency with the loss function.

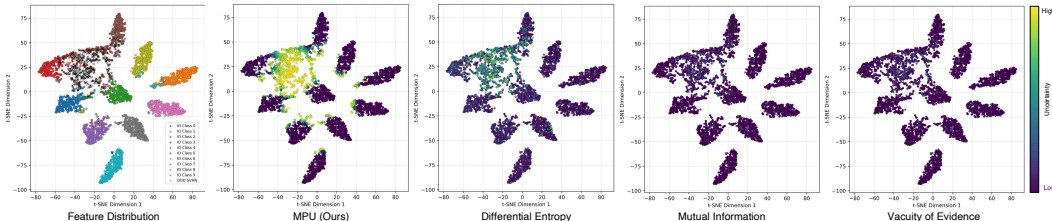

Figure 4: t-SNE visualization of feature distributions for ID (CIFAR-10) and OOD (SVHN) samples, along with their predicted uncertainties under different measures.

(3) While not always top-ranked for OOD detection, UCE+MPU is highly competitive, often achieving second-best performance. For in-distribution tasks, UCE with exponential activation delivers the highest accuracy, while MPU excels at misclassification detection. Notably, simply replacing VoE with MPU as the uncertainty measure boosts AUPR scores by 38.40, 22.47, 19.06, 26.69, and 34.14 across OOD datasets.

**UCE+MPU Becomes More Effective with Increasing Number of Classes.** We further benchmarked several state-of-the-art EDL methods on CIFAR-100. As shown in Table 3, we evaluated different methods combined with various uncertainty measures on both OOD detection and misclassification detection. Our results show that UCE+MPU consistently achieves the best performance on certain OOD datasets, including GTSRB, Places365, and CIFAR-10. In addition, MPU delivers the strongest results in misclassification detection on CIFAR-100, underscoring its effectiveness in more challenging, large-class classification scenarios.

**Uncertainty Ranking Exists Only within UCE Loss.** A consistent ranking of uncertainty measures exists when training with UCE loss: MPU > DE > MI > VoE. This ordering does not hold for other methods based on the Brier score loss (e.g. $\mathcal{I}$-EDL, R-EDL, and Re-EDL). Therefore, previous works typically report the performance of their methods across all uncertainty measures, but they cannot determine which uncertainty measure is optimal for their models.

**Visual Comparison of Uncertainty Measures.** To further understand the behavior of different uncertainty measures, we visualize the 2D feature distributions of ID and OOD samples using t-SNE, along with their predicted uncertainties from a model trained with UCE loss. As shown in Fig. 4, MPU assigns higher uncertainty to OOD and misclassified samples, with a clearer separation compared to other measures. Among them, Differential Entropy, which satisfies the optimization consistency property, performs well, while Mutual Information and Vacuity of Evidence show smaller uncertainty differences between ID and OOD samples. This visualization further highlights MPU's effectiveness in enhancing discernibility, particularly for OOD and misclassified instances.

## 6    CONCLUSION

**Summary.** In this work, we revisited the reliability of uncertainty measures in evidential deep learning (EDL) from an optimization perspective. We argued that a desirable uncertainty measure should be consistent with the model's optimization dynamics, i.e., samples closer to the global optimum should naturally exhibit lower uncertainty. Based on this principle, we proposed a new uncertainty measure tailored for UCE loss and demonstrated its effectiveness through extensive experiments. Our findings highlight the fundamental link between uncertainty quantification and optimization, suggesting that designing uncertainty measures with optimization consistency in mind can serve as a promising direction for future research in EDL and beyond.

**Deficiencies and Future directions.** Although our method yields strong results, it has notable limitations. On datasets with fewer classes (e.g., CIFAR-10), it underperforms Re-EDL trained with the Brier score, indicating that the UCE loss may not be universally optimal. This motivates exploring alternative losses such as the Brier score and examining how different losses and regularization terms (e.g., entropy minimization, KL divergence) jointly affect optimization and uncertainty estimation. Both theoretical and empirical studies in these directions are valuable avenues for future work.

## ACKNOWLEDGMENTS AND AUTHOR CONTRIBUTIONS

This work was supported by the National Natural Science Foundation of China (No. 62472315, 62476165).

## REPRODUCIBILITY STATEMENT

To ensure the reproducibility of our research, we provide comprehensive details of our methodology, experiments, and results. Detailed derivations and proofs for our theoretical claims are provided in the Appendix: objective functions are presented in Appendix A, uncertainty measures in Appendix B, the optimal solution of the SVM dual problem in Appendix C.1 and the margin-aware property of UCE loss in Appendix C.2. All code related to model training and feature visualization is included in the supplementary materials.

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

## THE USE OF LARGE LANGUAGE MODELS (LLMS)

We acknowledge the use of a Large Language Model (LLM) as a general-purpose writing assistant in preparing this manuscript. Its role was confined to improving grammar and language, and rephrasing sentences for clarity. All content was authored, and all LLM suggestions were critically reviewed and approved, by the human authors, who retain full responsibility for the final manuscript.

## A  DERIVATION OF THE OBJECTIVE FUNCTIONS

We present step-by-step analytical derivations of the UCE loss, the Brier score loss, and the Type-II maximum-likelihood loss. The derivation of the Brier score loss follows directly from the EDL (Sensoy et al., 2018), while the derivations of the UCE loss and the Type-II maximum-likelihood loss are original to this work and are provided in full detail.

### A.1  DERIVATION OF UNCERTAINTY-AWARE CROSS ENTROPY (UCE) LOSS

Consider a $K$-class classification problem, where the model predicts a Dirichlet distribution $\mathrm{Dir}(\boldsymbol{p}_i \,|\, \boldsymbol{\alpha}_i)$ over the categorical probabilities $\boldsymbol{p}_i = (p_{i1}, \ldots, p_{iK}) \in \Delta^{K-1}$, with parameters $\boldsymbol{\alpha}_i = (\alpha_{i1}, \ldots, \alpha_{iK})$. The Uncertainty-aware Cross-Entropy (UCE) loss is defined as the expected cross-entropy loss under the Dirichlet distribution:

$$\mathcal{L}_{\mathrm{UCE}}(\boldsymbol{\Theta}) = \int_{\boldsymbol{p}_i \in \Delta^{K-1}} \left[ -\sum_{j=1}^{K} y_{ij} \log(p_{ij}) \right] \mathrm{Dir}(\boldsymbol{p}_i \,|\, \boldsymbol{\alpha}_i) \, d\boldsymbol{p}_i, \tag{10}$$

where $\boldsymbol{y}_i$ is the one-hot label vector and the Dirichlet pdf is

$$\mathrm{Dir}(\boldsymbol{p}_i \,|\, \boldsymbol{\alpha}_i) = \frac{1}{B(\boldsymbol{\alpha}_i)} \prod_{j=1}^{K} p_{ij}^{\alpha_{ij}-1}, \quad B(\boldsymbol{\alpha}_i) = \int_{\Delta^{K-1}} \prod_{j=1}^{K} p_{ij}^{\alpha_{ij}-1} \, d\boldsymbol{p}_i \tag{11}$$

is the multivariate Beta function. Differentiating $B(\boldsymbol{\alpha})$ with respect to $\alpha_k$ and exchanging the order of differentiation and integration yields

$$\frac{\partial B(\boldsymbol{\alpha})}{\partial \alpha_k} = \int_{\Delta^{K-1}} \prod_{j=1}^{K} p_j^{\alpha_j-1} \log p_k \, d\boldsymbol{p}. \tag{12}$$

Dividing both sides by $B(\boldsymbol{\alpha})$, we obtain

$$\mathbb{E}_{\boldsymbol{p} \sim D(\boldsymbol{\alpha})}[\log(p_k)] = \int_{\Delta^{K-1}} \log p_k \cdot D(\boldsymbol{p} \,|\, \boldsymbol{\alpha}) \, d\boldsymbol{p} = \frac{1}{B(\boldsymbol{\alpha})} \frac{\partial B(\boldsymbol{\alpha})}{\partial \alpha_k}. \tag{13}$$

Comparing Eq. 13 with Eq. 12, we have

$$\mathbb{E}_{\boldsymbol{p} \sim D(\boldsymbol{\alpha})}[\log p_k] = \psi(\alpha_k) - \psi(S). \tag{14}$$

Finally, the loss can be written as the Dirichlet expectation of the negative log-likelihood:

$$\mathcal{L}_{\mathrm{UCE}}(\boldsymbol{\Theta}) = -\sum_{j=1}^{K} y_{ij} \, \mathbb{E}_{\boldsymbol{p} \sim D(\boldsymbol{\alpha}_i)}[\log p_{ij}], \tag{15}$$

which, upon substitution of the above identity, gives

$$\mathcal{L}_{\mathrm{UCE}}(\boldsymbol{\Theta}) = -\sum_{j=1}^{K} y_{ij} \big(\psi(\alpha_{ij}) - \psi(S_i)\big) = \sum_{j=1}^{K} y_{ij} \big(\psi(S_i) - \psi(\alpha_{ij})\big). \tag{16}$$

## A.2 Brier Score

The Brier Score loss takes the expected squares error loss between sampled probability $\boldsymbol{p}$ and label $\boldsymbol{y}$ as follows:

$$\mathcal{L}_{\text{Brier}}(\boldsymbol{\Theta}) = \int_{\boldsymbol{p}_i \in \Delta^{K-1}} \|\boldsymbol{y}_i - \boldsymbol{p}_i\|_2^2 \frac{1}{B(\boldsymbol{\alpha}_i)} \prod_{j=1}^K p_{ij}^{\alpha_{ij}-1} d\boldsymbol{p}_i$$

$$= \sum_{j=1}^K \mathbb{E}\left[ y_{ij}^2 - 2 y_{ij} p_{ij} + p_{ij}^2 \right] = \sum_{j=1}^K \left( y_{ij}^2 - 2 y_{ij} \mathbb{E}[p_{ij}] + \mathbb{E}[p_{ij}^2] \right)$$

Using the identity $\mathbb{E}[p_{ij}^2] = \mathbb{E}[p_{ij}]^2 + \text{Var}(p_{ij})$, we get:

$$\mathcal{L}_{\text{Brier}}(\boldsymbol{\Theta}) = \sum_{j=1}^K \left( (y_{ij} - \mathbb{E}[p_{ij}])^2 + \text{Var}(p_{ij}) \right)$$

$$= \sum_{j=1}^K \left( \underbrace{(y_{ij} - \alpha_{ij}/S_i)^2}_{\mathcal{L}_{ij}^{err}} + \underbrace{\frac{\alpha_{ij}(S_i - \alpha_{ij})}{S_i^2(S_i + 1)}}_{\mathcal{L}_{ij}^{var}} \right) \tag{17}$$

$$= \sum_{j=1}^K \left( (y_{ij} - \hat{p}_{ij})^2 + \frac{\hat{p}_{ij}(1 - \hat{p}_{ij})}{S_i + 1} \right).$$

$\mathcal{L}_{ij}^{err}$ is the bias term, which measures the squared difference between the predicted mean probability $\hat{p}_{ij}$ and the true label $y_{ij}$, reflecting the accuracy of the prediction. $\mathcal{L}_{ij}^{var}$ is the variance term, which captures the uncertainty of the prediction arising from the Dirichlet distribution over class probabilities, reflecting the model's confidence. Thus, the Brier Score loss naturally decomposes into an error term and an uncertainty term, providing a unified way to jointly account for prediction accuracy and epistemic uncertainty in evidential deep learning.

## A.3 Type II Maximum Likelihood Loss

The Type II Maximum Likelihood (T2ML) loss, also known as the marginal likelihood loss, is derived by maximizing the likelihood of the observed labels $\boldsymbol{y}_i$ under the predictive Dirichlet distribution parameterized by $\boldsymbol{\alpha}_i$. Intuitively, it encourages the model to assign higher Dirichlet concentration to the correct class, while naturally accounting for the uncertainty encoded in the distribution over class probabilities. Formally, the loss is defined as

$$\mathcal{L}_{\text{T2ML}}(\boldsymbol{\Theta}) = -\log \left( \int \prod_{j=1}^K p_{ij}^{y_{ij}} \cdot \frac{1}{B(\boldsymbol{\alpha}_i)} \prod_{j=1}^K p_{ij}^{\alpha_{ij}-1} d\boldsymbol{p}_i \right) = \sum_{j=1}^K y_{ij} \left( \log S_i - \log \alpha_{ij} \right), \tag{18}$$

Ignoring the sample index $i$ for simplicity, consider

$$\int_{\boldsymbol{p} \in \Delta^{K-1}} \prod_{j=1}^K p_j^{y_j} \cdot \frac{1}{B(\boldsymbol{\alpha})} \prod_{j=1}^K p_j^{\alpha_j - 1} d\boldsymbol{p}. \tag{19}$$

Since the integrand is proportional to a Dirichlet density with parameters $\boldsymbol{\alpha} + \boldsymbol{y}$, we can rewrite

$$\int_{\boldsymbol{p} \in \Delta^{K-1}} \frac{1}{B(\boldsymbol{\alpha})} \prod_{j=1}^K p_j^{y_j + \alpha_j - 1} d\boldsymbol{p} = \frac{B(\boldsymbol{\alpha} + \boldsymbol{y})}{B(\boldsymbol{\alpha})}. \tag{20}$$

Thus, the loss can be expressed as

$$\mathcal{L}_{\text{T2ML}}(\boldsymbol{\Theta}) = -\log \frac{B(\boldsymbol{\alpha} + \boldsymbol{y})}{B(\boldsymbol{\alpha})}. \tag{21}$$

Expanding the Beta functions in terms of Gamma functions gives

$$B(\boldsymbol{\alpha}) = \frac{\prod_{j=1}^{K} \Gamma(\alpha_j)}{\Gamma(S)}, \quad B(\boldsymbol{\alpha} + \boldsymbol{y}) = \frac{\prod_{j=1}^{K} \Gamma(\alpha_j + y_j)}{\Gamma(S+1)}. \tag{22}$$

Because $\boldsymbol{y}$ is one-hot, if $y_k = 1$ for the true class $k$ and $y_j = 0$ for $j \neq k$, we have

$$\frac{B(\boldsymbol{\alpha} + \boldsymbol{y})}{B(\boldsymbol{\alpha})} = \frac{\Gamma(\alpha_k + 1)}{\Gamma(\alpha_k)} \cdot \frac{\Gamma(S)}{\Gamma(S+1)}. \tag{23}$$

Using the Gamma identity $\Gamma(z+1) = z\,\Gamma(z)$, this simplifies to

$$\frac{B(\boldsymbol{\alpha} + \boldsymbol{y})}{B(\boldsymbol{\alpha})} = \frac{\alpha_c}{S}. \tag{24}$$

Therefore,

$$\mathcal{L}_{\text{T2ML}}(\boldsymbol{\Theta}) = -\log \frac{\alpha_k}{S} = \log S - \log \alpha_k. \tag{25}$$

Finally, reinstating the one-hot form of $\boldsymbol{y}$, we can write

$$\mathcal{L}_{\text{T2ML}}(\boldsymbol{\Theta}) = \sum_{j=1}^{K} y_{ij} \big( \log S_i - \log \alpha_{ij} \big) = \log S_i - \log \alpha_{ik}, \tag{26}$$

which completes the proof.

## B  DERIVATION OF THE UNCERTAINTY MEASURE

In this section, we provide detailed derivations of the uncertainty measures used in our work, including mutual information, differential entropy, and vacuity of evidence. These measures constitute fundamental background concepts, and their formulations have been summarized in prior works such as PriorNet (Malinin & Gales, 2018), EDL (Sensoy et al., 2018) and R-EDL (Chen et al., 2023b). We include them here for completeness and to maintain a self-contained presentation.

**Mutual Information.** A fundamental identity in information theory is that the Shannon entropy of a random variable $X$ can be additively decomposed into the mutual information between $X$ and $Y$, and the conditional entropy of $X$ given $Y$ (Ash, 2012):

$$\mathbb{H}(X) = \mathbb{I}(X;Y) + \mathbb{H}(X \mid Y) \tag{27}$$

Follow this idea, Malinin & Gales (2018) propose a method to explicitly model and decompose predictive total uncertainty into two components: *data uncertainty* (aleatoric uncertainty) and *distributional uncertainty* (epistemic uncertainty). The total uncertainty in the prediction is measured by the Shannon entropy of the expected categorical distribution conditioned

$$\mathbb{H}\left[p(y|\boldsymbol{p})\right] = \mathbb{E}_{\boldsymbol{p} \sim \text{Dir}(\boldsymbol{\alpha})}[p(y \mid \boldsymbol{p})] = -\sum_{j=1}^{K} \frac{\alpha_j}{S} \log \frac{\alpha_j}{S}, \tag{28}$$

Aleatoric uncertainty (or data uncertainty) corresponds to the expected entropy of the categorical distributions sampled from the Dirichlet prior, commonly referred to as the *conditional entropy*

$$\mathbb{E}_{\boldsymbol{p} \sim \text{Dir}(\boldsymbol{\alpha})} \left[\mathbb{H}[p(y \mid \boldsymbol{p})]\right] = \mathbb{E}_{\boldsymbol{p}} \left[ -\sum_{j=1}^{K} p_j \log p_j \right]$$

$$= -\sum_{j=1}^{K} \frac{\alpha_j}{S} \left(\psi(\alpha_j + 1) - \psi(S + 1)\right) \tag{29}$$

$$= \psi(S + 1) - \sum_{j=1}^{K} \frac{\alpha_j}{S} \psi(\alpha_j + 1)$$

where $\psi(\cdot)$ is the digamma function. For epistemic uncertainty, it is measured by the mutual information between predictions and the Dirichlet parameters as

$$\mathbb{I}(y, \boldsymbol{p}) = \mathbb{H}\left[\mathbb{E}_{\boldsymbol{p} \sim \text{Dir}(\boldsymbol{\alpha})}[p(y|\boldsymbol{p})]\right] - \mathbb{E}_{\boldsymbol{p} \sim \text{Dir}(\boldsymbol{\alpha})}[\mathbb{H}[p(y \mid \boldsymbol{p})]]. \tag{30}$$

This mutual information quantifies how much of the total uncertainty arises from uncertainty in the model parameters (i.e., distribution over categorical distributions), and thus reflects *epistemic uncertainty*. It is defined as

$$\underbrace{\mathbb{I}[y, \boldsymbol{p}]}_{\text{Epistemic Uncertainty}} \approx \underbrace{\mathbb{H}\left[\mathbb{E}_{\boldsymbol{p} \sim \text{Dir}(\boldsymbol{\alpha})}[p(y|\boldsymbol{p})]\right]}_{\text{Total Uncertainty}} - \underbrace{\mathbb{E}_{\boldsymbol{p} \sim \text{Dir}(\boldsymbol{\alpha})}[\mathbb{H}[p(y|\boldsymbol{p})]]}_{\text{Aleatoric Uncertainty}}$$

$$= -\sum_{j=1}^{K} \frac{\alpha_j}{S} \ln \frac{\alpha_j}{S} + \sum_{j=1}^{K} \frac{\alpha_j}{S}\left(\psi(\alpha_j + 1) - \psi(S + 1)\right) \tag{31}$$

$$= -\sum_{j=1}^{K} \frac{\alpha_j}{S}\left(\ln \frac{\alpha_j}{S} - \psi(\alpha_j + 1) + \psi(S + 1)\right).$$

**Differential Entropy.** Differential entropy is also a prevalent measure of epistemic uncertainty, where a lower value indicates that the model yields a sharper distribution, and a higher value means a more uniform Dirichlet distribution. It is defined as

$$\text{ENT}(\text{Dir}(\boldsymbol{p} \mid \boldsymbol{\alpha})) = -\int_{\boldsymbol{p} \in \Delta^{K-1}} \text{Dir}(\boldsymbol{p} \mid \boldsymbol{\alpha}) \log \text{Dir}(\boldsymbol{p} \mid \boldsymbol{\alpha}) \, d\boldsymbol{p}, \tag{32}$$

The closed-form expression is given by

$$\text{ENT}(\text{Dir}(\boldsymbol{p} \mid \boldsymbol{\alpha})) = \log B(\boldsymbol{\alpha}) + (S - K)\psi(S) - \sum_{j=1}^{K}(\alpha_j - 1)\psi(\alpha_j), \tag{33}$$

**Vacuity of Evidence.** EDL (Sensoy et al., 2018), RED (Pandey & Yu, 2023), $\mathcal{I}$-EDL (Deng et al., 2023), R-EDL (Chen et al., 2023b), and H-EDL (Qu et al., 2024), which are grounded in Subjective Logic (Jøsang, 2016) and Dempster-Shafer Theory (Dempster, 1968), represent uncertainty using evidence mass values. Subjective Logic (Jøsang, 2016) provides a principled framework for modeling predictive uncertainty by interpreting the output of a neural network as an *opinion*—a structured representation of uncertainty over a discrete set of classes. Unlike conventional classifiers that output categorical probabilities, EDL models produce non-negative evidence values $\boldsymbol{e} = [e_1, e_2, \ldots, e_K]$ for each of the $K$ classes. These evidence values parameterize a Dirichlet distribution $\text{Dir}(\boldsymbol{\alpha})$, where $\alpha_j = e_j + 1$. In Subjective Logic, an opinion over a finite domain is characterized by three components: the belief mass $b_j$, the base rate $a_j$, and the uncertainty mass $u$, satisfying:

$$b_j + u \cdot a_j = \mathbb{E}[p_j], \quad \text{and} \quad \sum_{j=1}^{K} b_j + u = 1 \tag{34}$$

where $p_j$ denotes the probability assigned to class $j$. These quantities relate to the Dirichlet parameters as follows: The belief mass $b_k$ is proportional to the evidence for class $k$:

$$b_k = \frac{e_k}{S}, \quad \text{where } S = \sum_{j=1}^{K}(e_j + 1) = \sum_{j=1}^{K} \alpha_j \tag{35}$$

The base rate $a_k$ is typically assumed to be uniform, i.e., $a_k = 1/K$. The uncertainty mass $u$ is defined as:

$$u = \frac{K}{\sum_{j=1}^{K} \alpha_j} = \frac{K}{S} \tag{36}$$

This uncertainty mass $u$ is referred to as vacuity of evidence in EDL literature, and it quantifies the degree of epistemic uncertainty due to a lack of evidence. When the total evidence is low (e.g., under out-of-distribution or ambiguous inputs), $S$ becomes small and $u$ approaches 1, indicating that the model abstains from committing belief to any specific class. Conversely, high total evidence yields a low vacuity, reflecting confident predictions based on strong feature-based support. This opinion-based interpretation highlights the epistemic nature of uncertainty in EDL and differentiates it from aleatoric uncertainty captured by distributional spread in conventional probabilistic models.

## C    PROOF OF THE MARGIN-BASED THEOREM AND PROPOSITION

### C.1    DERIVATION OF OPTIMAL SOLUTION FOR C&S MULTI-CLASS SVM

**Definition 1.** The primal optimization problem for the Crammer and Singer multi-class SVM (Crammer & Singer, 2001) is defined as

$$\min_{\boldsymbol{w}, \boldsymbol{\xi}} \frac{1}{2}\beta \sum_{j=1}^{K} \boldsymbol{w}_j^\top \boldsymbol{w}_j + \sum_{i=1}^{N} \xi_i \tag{37}$$

$$\text{s.t.} \quad \boldsymbol{w}_{y_i}^\top \boldsymbol{x}_i - \boldsymbol{w}_j^\top \boldsymbol{x}_i + \delta_{y_i,j} \geq 1 - \xi_i, \quad \forall i \in [1, N], j \in [1, K]$$

where $\beta > 0$ is a regularization constant, and $N$ is the number of training samples. The $N$ slack variables $\xi_{i \in [N]}$ quantify the degree of classification error. $\delta$ is the Kronecker delta function $\delta_{p,q} = 1$ if $p = q$ else 0 otherwise. For $j = y_i$, the inequality constraints become $\xi_i \geq 0$. For each sample $\boldsymbol{x}_i$, $\xi_i$ measures the minimal adjustment needed for the model to satisfy the classification constraint. If a data point lies outside the margin or is correctly classified, then $\xi_i = 0$; otherwise, $\xi_i > 0$, indicating that the sample is either misclassified or within the incorrect classification margin. By minimizing the sum of these slack variables and weight norm, Crammer and Singer's $K$-class SVM seeks a balance between accurate classification and model complexity. The primal optimization problem in Eq. 37 can be solved by applying the Karush-Kuhn-Tucker (KKT) theorem (Bertsekas, 1997), which provides the necessary conditions for optimality in constrained optimization problems. By introducing a set of dual variables $\boldsymbol{\eta}$ for the constraints, we construct the Lagrangian function and derive the corresponding dual optimization problem:

$$\max_{\boldsymbol{\eta}} \min_{\boldsymbol{w}, \boldsymbol{\xi}} \mathcal{L}(\boldsymbol{w}, \boldsymbol{\xi}, \boldsymbol{\eta}) \quad \text{s.t.} \quad \eta_{i,j} \geq 0, \quad \forall i \in [1, N], j \in [1, K] \tag{38}$$

where

$$\mathcal{L}(\boldsymbol{w}, \boldsymbol{\xi}, \boldsymbol{\eta}) = \frac{1}{2}\beta \sum_{j=1}^{K} \boldsymbol{w}_j^\top \boldsymbol{w}_j + \sum_{i=1}^{N} \xi_i + \sum_{i=1}^{N}\sum_{j=1}^{K} \eta_{i,j} \left[\boldsymbol{w}_j^\top x_i - \boldsymbol{w}_{y_i}^\top x_i - \delta_{y_i,j} + 1 - \xi_i \right] \tag{39}$$

We can solve for optimal $\boldsymbol{w}_j$ as function of $\eta$ by finding the saddle point by differentiating $\boldsymbol{w}_j$ and $\xi_i$, we require

$$\frac{\partial \mathcal{L}}{\partial \xi_i} = 1 - \sum_{j=1}^{K} \eta_{i,j} = 0 \tag{40}$$

Then, we get $\sum_{q=1}^{K} \eta_{i,q} = 1, \forall i \in [N]$. Similarly, for $\boldsymbol{w}_j$, we require

$$\frac{\partial \mathcal{L}}{\partial \boldsymbol{w}_j} = \beta \boldsymbol{w}_j + \sum_{i=1}^{N} \eta_{i,j} \boldsymbol{x}_i - \sum_{i=1}^{N} \delta_{y_i,j} \left(\sum_{q=1}^{K} \eta_{i,q}\right) \boldsymbol{x}_i \tag{41}$$

$$= \sum_{i=1}^{N} \eta_{i,j} \boldsymbol{x}_i - \sum_{i=1}^{N} \delta_{y_i,j} \boldsymbol{x}_i + \beta \boldsymbol{w}_j = 0 \tag{42}$$

Finally, the optimal form of $\boldsymbol{w}_j$ becomes:

$$\boldsymbol{w}_j = \beta^{-1} \left[\sum_{i=1}^{n} \tau_{ij} \boldsymbol{x}_i \right] \tag{43}$$

where $\boldsymbol{\tau}_i = \boldsymbol{y}_i - \boldsymbol{\eta}_i$ denotes the difference between the one-hot label distribution $\boldsymbol{y}_i$ concentrating on the ground-truth label and the distribution $\boldsymbol{\eta}_i$ obtained by the optimization problem. In Eq. 43, the optimal $\boldsymbol{w}_j$ is expressed as a linear combination of $\boldsymbol{x}_{i \in [N]}$, with weights given by $\tau_{ij}$. Since for $\boldsymbol{x}_i$, the constraints $\eta_{i,1}, \ldots, \eta_{i,K} \geq 0$ and $\sum_{j=1}^{K} \eta_{i,j} = 1, \forall i \in [N]$ hold, the composition of $\boldsymbol{w}_j$ can be interpreted probabilistically. Specifically, $\boldsymbol{x}_i$ is considered a support pattern if and only if its corresponding distribution is not concentrated on the correct label $\boldsymbol{y}_i$. Therefore, the classifiers $\boldsymbol{w}_j$ for $j \in [K]$ are constructed using patterns whose distribution $\boldsymbol{\tau}$ deviate from the true label information (i.e. whose labels are uncertain (Crammer & Singer, 2001)).

## C.2 DERIVATION OF MARGIN-AWARE PROPERTY OF EDL LOSS

Before proceeding with the proof, we first introduce Lemma 1.

**Lemma 1.** For any $b > a > 0$, it holds that $\psi(b) - \psi(a) \geq \log(b) - \log(a)$.

**Proof of Lemma 1.** Define the auxiliary function $h(x) = \psi(x) - \log(x)$ for $x > 0$. To prove the claim, it suffices to show that $h(b) \geq h(a)$ whenever $b > a$, which is equivalent to showing that $h(x)$ is a monotonically non-decreasing function. This can be verified by examining its derivative $h'(x)$:

$$h'(x) = \frac{d}{dx}(\psi(x) - \log(x)) = \psi'(x) - \frac{1}{x}. \tag{44}$$

Here, $\psi'(x)$ is the trigamma function. A standard integral representation of the trigamma function is

$$\psi'(x) = \int_0^\infty \frac{te^{-xt}}{1 - e^{-t}} \, dt. \tag{45}$$

A well-known inequality for the exponential function is $1 - e^{-t} \leq t$ for all $t \geq 0$. Since both sides are positive for $t > 0$, taking reciprocals reverses the inequality:

$$\frac{1}{1 - e^{-t}} \geq \frac{1}{t}. \tag{46}$$

Substituting this bound into the integral for $\psi'(x)$ gives

$$\psi'(x) \geq \int_0^\infty \frac{te^{-xt}}{t} \, dt = \int_0^\infty e^{-xt} \, dt = \left[ -\frac{1}{x} e^{-xt} \right]_0^\infty = \frac{1}{x}. \tag{47}$$

Thus, $\psi'(x) \geq \frac{1}{x}$, which implies

$$h'(x) = \psi'(x) - \frac{1}{x} \geq 0. \tag{48}$$

Since its derivative is non-negative, the function $h(x)$ is monotonically non-decreasing.

Let $y$ denote the ground-truth label. Since $S > \alpha_y$ (as $S$ contains $\alpha_y$ plus other positive terms), it follows that $h(S) \geq h(\alpha_y)$. That is,

$$\psi(S) - \log(S) \geq \psi(\alpha_y) - \log(\alpha_y). \tag{49}$$

Rearranging this inequality yields the desired lower bound for the loss function:

$$\mathcal{L}_{\text{UCE}}(\mathbf{\Theta}) = \psi(S) - \psi(\alpha_y) \geq \log(S) - \log(\alpha_y) = \log\left(\frac{S}{\alpha_y}\right). \tag{50}$$

**Theorem 1. The Margin-Aware Property of UCE Loss.** It can be shown that the UCE loss $\mathcal{L}_{\text{UCE}}$ can be lower-bounded in the following form:

$$\mathcal{L}_{\text{UCE}}(\boldsymbol{x}, y, \boldsymbol{W}, \boldsymbol{\Psi}) \geq \phi\left(\sum_{j \neq y} \left(\boldsymbol{w}_j^\top \boldsymbol{\Psi}(\boldsymbol{x}) - \boldsymbol{w}_y^\top \boldsymbol{\Psi}(\boldsymbol{x})\right)\right) = \phi\left(-\mathcal{M}(\boldsymbol{x}, y; \boldsymbol{W}, \boldsymbol{\Psi})\right) \tag{51}$$

where $\phi(t) = \log\left(1 + (K - 1)\min\left(1, \exp\left(t\right)\right)\right)$, and the inequality achieves equality when $z_j = z_{j'}$ for all $j, j' \neq y$, and $\mathcal{M}(\boldsymbol{x}, y; \boldsymbol{W}, \boldsymbol{\Psi}) = \sum_{j \neq y} \left(\boldsymbol{w}_y^\top \boldsymbol{\Psi}(\boldsymbol{x}) - \boldsymbol{w}_j^\top \boldsymbol{\Psi}(\boldsymbol{x})\right)$.

**Proof of Theorem 1.** First, we transform the expression into has a lower bound with the form $\log(S) - \log(\alpha_y)$ with Lemma 1, which is lower-bounded by the logarithm of the ratio $S/\alpha_y$. Now, let's analyze the argument of the logarithm, $\frac{S}{\alpha_y}$.

$$\frac{S}{\alpha_y} = \frac{\sum_{j=1}^K \alpha_j}{\alpha_y} = \frac{\alpha_y + \sum_{j \neq y} \alpha_j}{\alpha_y} = 1 + \frac{\sum_{j \neq y} \alpha_j}{\alpha_y} \tag{52}$$

So our lower bound becomes:

$$\mathcal{L}_{\text{UCE}}(\boldsymbol{\Theta}) \geq \log\left(1 + \frac{\sum_{j \neq k} \alpha_j}{\alpha_k}\right) \tag{53}$$

We adopt the exponential activation function, i.e., $\sigma(z) = \exp(z)$ and $\alpha_j = \exp(z_j) + 1$. Then, the ratio term becomes:

$$\frac{\sum_{j \neq y} \alpha_j}{\alpha_y} = \frac{\sum_{j \neq y}(\exp(z_j) + 1)}{\exp(z_y) + 1} = \frac{(\sum_{j \neq y}\exp(z_j)) + (K - 1)}{\exp(z_y) + 1} \tag{54}$$

We can rewrite $\sum_{j \neq y}\exp(z_j)$ as

$$\sum_{j \neq y}\exp(z_j) = \sum_{j \neq y}\exp((z_j - z_y) + z_y) = \exp(z_y)\sum_{j \neq y}\exp(z_j - z_y). \tag{55}$$

For a convex function $\phi$ (here, $\phi(x) = e^x$), we have the Jensen's inequality that:

$$\frac{1}{n}\sum_{i=1}^{n}\phi(x_i) \geq \phi\left(\frac{1}{n}\sum_{i=1}^{n}x_i\right) \tag{56}$$

Applying this to our sum over the $K - 1$ non-target classes due to the convexity of exponential function, we have

$$\frac{1}{K-1}\sum_{j \neq y}\exp(z_j - z_y) \geq \exp\left(\frac{1}{K-1}\sum_{j \neq y}(z_j - z_y)\right) \tag{57}$$

Let's define the $M = \sum_{j \neq y}(z_j - z_y)$. The inequality becomes:

$$\sum_{j \neq k}\exp(z_j - z_y) \geq (K - 1)\exp\left(\frac{M}{K-1}\right) \tag{58}$$

Now, substitute Eq. 58 back into Eq. 55, we have

$$\sum_{j \neq k}\exp(z_j) = \exp(z_y)\sum_{j \neq y}\exp(z_j - z_y) \geq \exp(z_y)(K - 1)\exp\left(\frac{M}{K-1}\right) \tag{59}$$

Plugging this result back into Eq. 54, we have

$$\frac{\sum_{j \neq y} \alpha_j}{\alpha_y} = \frac{(\sum_{j \neq y}\exp(z_j)) + (K - 1)}{\exp(z_y) + 1} \geq \frac{\exp(z_k)(K - 1)\exp\left(\frac{M}{K-1}\right) + (K - 1)}{\exp(z_y) + 1}$$
$$= (K - 1)\frac{\exp(z_y)\exp\left(\frac{M}{K-1}\right) + 1}{\exp(z_y) + 1} \tag{60}$$

This expression still depends on $z_y$. To obtain a bound of the form $f(M)$, we must find a lower bound for this expression that is independent of $z_y$. Let's introduce the term

$$g(z_y, M) = \frac{\exp(z_y)\exp\left(\frac{M}{K-1}\right) + 1}{\exp(z_y) + 1}. \tag{61}$$

Let $u = \exp(z_k)$ and $C = \exp(\frac{M}{K-1})$. The expression is $\frac{uC+1}{u+1}$. The derivative with respect to $u$ is $\frac{C-1}{(u+1)^2}$. If $M > 0$, then $C > 1$, the derivative is positive, and the function is increasing in $u$ (and thus $z_y$). Its minimum value is approached as $z_y \to -\infty$, which is 1. If $M < 0$, then $C < 1$, the derivative is negative, and the function is decreasing in $u$ (and thus $z_y$). Its minimum value is approached as $z_y \to \infty$, which is $C = \exp(\frac{M}{K-1})$. As $z_y \to \infty$ (i.e., $u \to \infty$), the term approaches

$\frac{uC}{u} = C = \exp\left(\frac{M}{K-1}\right)$. As $z_y \to -\infty$ (i.e., $u \to 0$), the term approaches $\frac{1}{1} = 1$. Combining these two cases, we can establish a uniform lower bound:

$$\frac{\exp(z_y)\exp\left(\frac{M}{K-1}\right) + 1}{\exp(z_y) + 1} \geq \min\left(1, \exp\left(\frac{M}{K-1}\right)\right) \tag{62}$$

Therefore, we have a lower bound for our ratio that depends only on the margin $M$:

$$\frac{\sum_{j\neq y}\alpha_j}{\alpha_y} \geq (K-1)\min\left(1, \exp\left(\frac{M}{K-1}\right)\right) \tag{63}$$

Then, we get the final lower bound for the loss function:

$$\mathcal{L}_{\text{UCE}}(\boldsymbol{\Theta}) \geq \log\left(1 + \frac{\sum_{j\neq y}\alpha_j}{\alpha_y}\right) \geq \log\left(1 + (K-1)\min\left(1, \exp\left(\frac{\sum_{j\neq y}(z_j - z_y)}{K-1}\right)\right)\right) \tag{64}$$

Let $\phi(t) = \log\left(1 + (K-1)\min\left(1, \exp\left(t\right)\right)\right)$. It is straightforward to verify that $\phi(t)$ is a non-decreasing function of $t$ by computing its first derivative. Therefore, we have rigorously shown that the UCE loss function $\mathcal{L}_{\text{UCE}}(\boldsymbol{\Theta})$ admits a lower bound of the form $\phi\left(\sum_{j\neq y}(z_j - z_y)\right)$, where $\phi$ is a non-decreasing function. This implies that minimizing the UCE loss inherently promotes increasing the margin between the logit of the true class and those of the incorrect classes, thereby aligning with a desirable property of classification losses.

### C.3 GRADIENT ANALYSIS FOR THE CLASSIFIER VECTORS WITH UCE LOSS

**Proof of Proposition 1.** We aim to show that when optimizing an EDL classifier $\boldsymbol{W}$ with the UCE loss, the gradient update contributed by each sample is analogous to that in the dual problem of C&S SVMs, where it is expressed as a linear combination of the sample features and uncertain samples contribute more strongly to the update as

$$\nabla_{\boldsymbol{w}_j}\mathcal{L}_{\text{UCE}}(\boldsymbol{W}, \boldsymbol{\Psi}) \approx \sum_{i=1}^{N}\left(\delta_{y_i,j} - b_{ij}\right)\boldsymbol{\Psi}(\boldsymbol{x}_i), \quad b_{ij} = \frac{\alpha_{ij}-1}{S_i}. \tag{65}$$

The gradient descent update uses the negative gradient, $-\nabla_{\boldsymbol{w}_j}\mathcal{L}$. Suppose that the final gradient update is given by the accumulated contributions of all $N$ samples, we have

$$-\nabla_{\boldsymbol{w}_j}\mathcal{L}_{\text{UCE}}(\boldsymbol{w}_j) = -\sum_{i=1}^{N}\frac{\partial\mathcal{L}_i}{\partial\boldsymbol{w}_j} = \sum_{i=1}^{N}\left(\delta_{y_i,j}\psi_1(\alpha_{ij}) - \psi_1(S_i)\right)\frac{\partial\alpha_{ij}}{\partial z_{ij}}\boldsymbol{\Psi}(\boldsymbol{x}_i)$$

$$= \sum_{i=1,y_i=j}^{N}\left(\psi_1\left(\alpha_{ij}\right)\frac{\partial\alpha_{ij}}{\partial z_{ij}} - \psi_1\left(S_i\right)\frac{\partial\alpha_{ij}}{\partial z_{ij}}\right)\boldsymbol{\Psi}(\boldsymbol{x}_i) + \sum_{i=1,y_i\neq j}^{N}\left(-\psi_1\left(S_i\right)\frac{\partial\alpha_{ij}}{\partial z_{ij}}\right)\boldsymbol{\Psi}(\boldsymbol{x}_i) \tag{66}$$

where $\psi_1(z) = \frac{d}{dz}\ln\psi(z)$ denotes the trigamma function, which is strictly decreasing and convex on $(0, \infty)$, and $\delta_{y_i,j}$ is the Kronecker delta. When $z \gg 1$, we have $\psi_1(z) \approx \frac{1}{z}$. If we assume that $\alpha_{i,y_i} \gg 1$ and $S_i \gg 1$, this is typically easy to achieve in practice, since we use an exponential activation function, and the Dirichlet concentration on the correct class is usually much larger than the prior of 1. Consider the two terms that contribute to the classifier $\boldsymbol{w}_j$:

(1) The sum over samples with $y_i = j$:

$$\sum_{i:y_i=j}\left(\psi_1(\alpha_{ij})\frac{\partial\alpha_{ij}}{\partial z_{ij}} - \psi_1(S_i)\frac{\partial\alpha_{ij}}{\partial z_{ij}}\right)\boldsymbol{\Psi}(\boldsymbol{x}_i). \tag{67}$$

Substitute $\partial\alpha_{ij}/\partial z_{ij} = \alpha_{ij} - 1$, we have

$$\sum_{i:y_i=j}\left(\psi_1(\alpha_{ij}) - \psi_1(S_i)\right)(\alpha_{ij} - 1)\boldsymbol{\Psi}(\boldsymbol{x}_i). \tag{68}$$

Then apply the approximations (as $\psi_1(\alpha_{ij}) \approx \frac{1}{\alpha_{ij}}$ for $\alpha_{ij} \gg 1$), we have

$$\psi_1(\alpha_{ij})\,(\alpha_{ij} - 1) \approx \frac{\alpha_{ij} - 1}{\alpha_{ij}}, \qquad \psi_1(S_i)\,(\alpha_{ij} - 1) \approx \frac{\alpha_{ij} - 1}{S_i}, \tag{69}$$

then

$$\sum_{i:y_i=j} \Big( \psi_1(\alpha_{ij}) - \psi_1(S_i) \Big)(\alpha_{ij} - 1)\,\mathbf{\Psi}(\boldsymbol{x}_i) \approx \sum_{i:y_i=j} \left( \frac{\alpha_{ij} - 1}{\alpha_{ij}} - \frac{\alpha_{ij} - 1}{S_i} \right)\mathbf{\Psi}(\boldsymbol{x}_i). \tag{70}$$

(2) The sum over samples with $y_i \neq j$:

$$\sum_{i:y_i \neq j} \left( -\psi_1(S_i)\frac{\partial \alpha_{ij}}{\partial z_{ij}} \right)\mathbf{\Psi}(\boldsymbol{x}_i). \tag{71}$$

Substituting $\partial\alpha_{ij}/\partial z_{ij} = \alpha_{ij} - 1$ and using the same approximation for $\psi_1(S_i)(\alpha_{ij} - 1)$,

$$\sum_{i:y_i \neq j} \Big( -\psi_1(S_i)(\alpha_{ij} - 1) \Big)\mathbf{\Psi}(\boldsymbol{x}_i) \approx \sum_{i:y_i \neq j} \left( -\frac{\alpha_{ij} - 1}{S_i} \right)\mathbf{\Psi}(\boldsymbol{x}_i). \tag{72}$$

Combine the two sums and write both cases in a single sum over all $i$. First, observe that for the $y_i = j$ case the coefficient (after approximation) is

$$\frac{\alpha_{ij} - 1}{\alpha_{ij}} - \frac{\alpha_{ij} - 1}{S_i} \approx 1 - \frac{\alpha_{ij} - 1}{S_i}, \tag{73}$$

with the assumption that $\alpha_{ij} \gg 1$ for $y_i = j$. And for the $y_i \neq j$ case it is

$$-\frac{\alpha_{ij} - 1}{S_i}. \tag{74}$$

Define the scalar $b_{ij}$ by

$$b_{ij} \;=\; \frac{\alpha_{ij} - 1}{S_i}. \tag{75}$$

as in Subjective Logic (Jøsang, 2016), which is the belief mass that supports the $j$-th singleton. Hence both contributions can be written as the single sum for $\boldsymbol{w}_j$

$$\nabla_{\boldsymbol{w}_j}\mathcal{L}_{\mathrm{UCE}}(\boldsymbol{W}, \mathbf{\Psi}) \approx \sum_{i=1}^{N} \Big( \delta_{y_i,j} - b_{ij} \Big)\mathbf{\Psi}(\boldsymbol{x}_i). \tag{76}$$

## C.4 PROOF OF OPTIMIZATION CONSISTENCY FOR DIFFERENTIAL ENTROPY

**Proposition 2. Optimization-Consistency of Differential Entropy.** Let $\mathrm{ENT}(\mathrm{Dir}(\boldsymbol{p} \mid \boldsymbol{\alpha}))$ denote the differential entropy of the Dirichlet distribution parameterized by $\boldsymbol{\alpha}$ as

$$\begin{aligned}
\mathrm{ENT}(\mathrm{Dir}(\boldsymbol{p} \mid \boldsymbol{\alpha})) &= -\int_{\boldsymbol{p} \in \Delta^{K-1}} \mathrm{Dir}(\boldsymbol{p} \mid \boldsymbol{\alpha}) \log \mathrm{Dir}(\boldsymbol{p} \mid \boldsymbol{\alpha})\, d\boldsymbol{p} \\
&= \log B(\boldsymbol{\alpha}) + (S - K)\psi(S) - \sum_{j=1}^{K}(\alpha_j - 1)\psi(\alpha_j).
\end{aligned} \tag{77}$$

where $B(\boldsymbol{\alpha}) = \frac{\prod_{j=1}^{K}\Gamma(\alpha_j)}{\Gamma(S)}$. Then the uncertainty measure defined by differential entropy is *optimization-consistent* with respect to the UCE loss in the sense of Theorem 2. That is, for any two training samples $(\boldsymbol{x}, y)$ and $(\boldsymbol{x}', y')$, we have

$$\mathcal{L}_{\mathrm{UCE}}(\boldsymbol{x}, y, \boldsymbol{W}, \mathbf{\Psi}) \leq \mathcal{L}_{\mathrm{UCE}}(\boldsymbol{x}', y', \boldsymbol{W}, \mathbf{\Psi}) \implies \mathrm{ENT}(\mathrm{Dir}(\boldsymbol{p} \mid \boldsymbol{\alpha}(\boldsymbol{x}))) \leq \mathrm{ENT}(\mathrm{Dir}(\boldsymbol{p} \mid \boldsymbol{\alpha}(\boldsymbol{x}'))). \tag{78}$$

**Proof of Proposition 2.** From Theorem 4.2, the UCE loss for a sample $(\boldsymbol{x}, y)$ satisfies

$$\mathcal{L}_{\mathrm{UCE}}(\boldsymbol{x}, y) \geq \phi\Big( -\mathcal{M}(\boldsymbol{x}, y) \Big), \quad \mathcal{M}(\boldsymbol{x}, y) := \sum_{j \neq y}(z_y - z_j), \tag{79}$$

where $z_j = \boldsymbol{w}_j^\top \boldsymbol{\Psi}(\boldsymbol{x})$ and $\phi(t) = \log\left(1 + (K-1)\min(1, \exp(t))\right)$. By the monotonicity of $\phi$, for any two samples $(\boldsymbol{x}, y)$ and $(\boldsymbol{x}', y')$:

$$\mathcal{L}_{\text{UCE}}(\boldsymbol{x}, y) \leq \mathcal{L}_{\text{UCE}}(\boldsymbol{x}', y') \quad \Rightarrow \quad \mathcal{M}(\boldsymbol{x}, y) \geq \mathcal{M}(\boldsymbol{x}', y'). \tag{80}$$

Thus, a smaller UCE loss corresponds to a larger logit margin. Then, consider the differential entropy $\mathcal{H}(\boldsymbol{\alpha}) := \text{ENT}(\text{Dir}(\boldsymbol{p} \mid \boldsymbol{\alpha}))$ with the parameterization $\alpha_j = \exp(z_j) + 1$. Then the gradient with respect to $z_j$ for $\forall j \in [K]$ is

$$\frac{\partial \mathcal{H}}{\partial z_j} = \frac{\partial \mathcal{H}}{\partial \alpha_j}(\alpha_j - 1). \tag{81}$$

As

$$\frac{\partial}{\partial \alpha_j} \log B(\boldsymbol{\alpha}) = \psi(\alpha_j) - \psi(S), \tag{82}$$

$$\frac{\partial}{\partial \alpha_j}\left[(S - K)\psi(S)\right] = \psi(S) + (S - K)\psi_1(S), \tag{83}$$

$$\frac{\partial}{\partial \alpha_j}\left[-(\alpha_j - 1)\psi(\alpha_j)\right] = -\psi(\alpha_j) - (\alpha_j - 1)\psi_1(\alpha_j), \tag{84}$$

where $\psi_1(\cdot)$ is the trigamma function, satisfying $\psi_1(x) > 0$ and strictly decreasing for $x > 0$. Then, we have

$$\frac{\partial \mathcal{H}}{\partial \alpha_j} = (S - K)\psi_1(S) - (\alpha_j - 1)\psi_1(\alpha_j), \tag{85}$$

and

$$\frac{\partial \mathcal{H}}{\partial z_j} = \left((S - K)\psi_1(S) - (\alpha_j - 1)\psi_1(\alpha_j)\right)(\alpha_j - 1). \tag{86}$$

Since $\alpha_j = \exp(z_j) + 1 > 1$, we have $(\alpha_j - 1) > 0$, and the sign of $\frac{\partial \mathcal{H}}{\partial z_j}$ is determined by the expression:

$$A_j = (S - K)\psi_1(S) - (\alpha_j - 1)\psi_1(\alpha_j). \tag{87}$$

Consider the function $h(x) = (x - 1)\psi_1(x)$ for $x > 1$. The function $h(x)$ is monotonically increasing, and as $x \to \infty$, $h(x) \to 1$. We analyze the sign of $A_j$:

- For the true class $y$: When the margin $\mathcal{M}$ increases, $z_y$ increases, leading to an increase in $\alpha_y$. Since $S = \sum_{j=1}^K \alpha_j$ and the margin increase typically implies $\alpha_y$ dominates the sum, we have $S \approx \alpha_y$. Then:

$$A_y \approx (\alpha_y - K)\psi_1(\alpha_y) - (\alpha_y - 1)\psi_1(\alpha_y) = (1 - K)\psi_1(\alpha_y) < 0, \tag{88}$$

  since $K \geq 2$ and $\psi_1(\alpha_y) > 0$. Therefore, $\frac{\partial \mathcal{H}}{\partial z_y} < 0$, meaning that increasing $z_y$ decreases $\mathcal{H}$.

- For incorrect classes $j \neq y$: When the margin $\mathcal{M}$ increases, $z_j$ decreases, leading to $\alpha_j$ decreasing toward 1. As $\alpha_j \to 1$, $(\alpha_j - 1)\psi_1(\alpha_j) \to 0$, while $(S - K)\psi_1(S) > 0$ (since $S > K$), so $A_j > 0$. Therefore, $\frac{\partial \mathcal{H}}{\partial z_j} > 0$, meaning that decreasing $z_j$ decreases $\mathcal{H}$.

Thus, when the margin $\mathcal{M}$ increases (i.e., $z_y$ increases or $z_j$ decreases for $j \neq y$), the differential entropy $\mathcal{H}$ decreases. By Theorem 1, a smaller UCE loss implies a larger margin, which in turn implies a smaller differential entropy. Therefore:

$$\begin{aligned}\mathcal{L}_{\text{UCE}}(\boldsymbol{x}, y, \boldsymbol{W}, \boldsymbol{\Psi}) &\leq \mathcal{L}_{\text{UCE}}(\boldsymbol{x}', y', \boldsymbol{W}, \boldsymbol{\Psi}) \\ &\Longrightarrow \mathcal{M}(\boldsymbol{x}, y) \geq \mathcal{M}(\boldsymbol{x}', y') \\ &\Longrightarrow \mathcal{H}(\boldsymbol{\alpha}(\boldsymbol{x})) \leq \mathcal{H}(\boldsymbol{\alpha}(\boldsymbol{x}')),\end{aligned} \tag{89}$$

which completes the proof of optimization consistency.

Table 4: Performance comparison evaluated by classification accuracy and AUPR scores for misclassification detection on CIFAR-10 and CIFAR-10-C with different level of noise settings.

| Method | CIFAR-10 | | | | | CIFAR-10-C Level 1 | | | | |
| --- | --- | --- | --- | --- | --- | --- | --- | --- | --- | --- |
| | Cls Acc | MPU | DE | MI | VoE | Cls Acc | MPU | DE | MI | VoE |
| EDL | $88.48_{\pm0.32}$ | $98.74_{\pm0.08}$ | $98.74_{\pm0.07}$ | $98.74_{\pm0.08}$ | $98.74_{\pm0.08}$ | $84.64_{\pm0.34}$ | $97.95_{\pm0.11}$ | $97.95_{\pm0.11}$ | $97.94_{\pm0.11}$ | $97.94_{\pm0.11}$ |
| $\mathcal{I}$-EDL | $88.38_{\pm0.15}$ | $98.71_{\pm0.12}$ | $98.71_{\pm0.11}$ | $98.70_{\pm0.12}$ | $98.70_{\pm0.12}$ | $84.01_{\pm0.47}$ | $97.77_{\pm0.18}$ | $97.77_{\pm0.18}$ | $97.75_{\pm0.18}$ | $97.75_{\pm0.18}$ |
| R-EDL | $90.09_{\pm0.31}$ | $98.98_{\pm0.06}$ | $98.95_{\pm0.06}$ | $98.98_{\pm0.05}$ | $98.98_{\pm0.06}$ | $86.15_{\pm0.32}$ | $98.17_{\pm0.05}$ | $98.17_{\pm0.05}$ | $98.17_{\pm0.05}$ | $98.17_{\pm0.05}$ |
| Re-EDL | $90.13_{\pm0.25}$ | $98.76_{\pm0.06}$ | $98.77_{\pm0.07}$ | $98.69_{\pm0.08}$ | $98.81_{\pm0.05}$ | $86.34_{\pm0.41}$ | $97.93_{\pm0.13}$ | $97.92_{\pm0.13}$ | $97.83_{\pm0.14}$ | $97.81_{\pm0.15}$ |
| Our | $\mathbf{93.41_{\pm0.24}}$ | $\mathbf{99.43_{\pm0.01}}$ | $99.34_{\pm0.04}$ | $99.12_{\pm0.07}$ | $96.66_{\pm0.36}$ | $\mathbf{89.26_{\pm0.10}}$ | $\mathbf{98.70_{\pm0.03}}$ | $98.59_{\pm0.04}$ | $98.32_{\pm0.11}$ | $94.79_{\pm0.56}$ |

| Method | CIFAR-10-C Level 2 | | | | | CIFAR-10-C Level 3 | | | | |
| --- | --- | --- | --- | --- | --- | --- | --- | --- | --- | --- |
| | Acc | MPU | DE | MI | VoE | Acc | MPU | DE | MI | VoE |
| EDL | $80.47_{\pm0.40}$ | $96.79_{\pm0.17}$ | $96.79_{\pm0.17}$ | $96.78_{\pm0.17}$ | $96.78_{\pm0.17}$ | $76.08_{\pm0.61}$ | $95.35_{\pm0.30}$ | $95.35_{\pm0.30}$ | $95.33_{\pm0.30}$ | $95.33_{\pm0.30}$ |
| $\mathcal{I}$-EDL | $79.29_{\pm0.82}$ | $96.42_{\pm0.30}$ | $96.42_{\pm0.30}$ | $96.38_{\pm0.30}$ | $96.38_{\pm0.30}$ | $74.67_{\pm1.03}$ | $94.72_{\pm0.51}$ | $94.72_{\pm0.51}$ | $94.67_{\pm0.51}$ | $94.66_{\pm0.51}$ |
| R-EDL | $81.73_{\pm0.35}$ | $96.99_{\pm0.14}$ | $96.99_{\pm0.14}$ | $96.99_{\pm0.14}$ | $96.99_{\pm0.14}$ | $77.24_{\pm0.52}$ | $95.51_{\pm0.45}$ | $95.51_{\pm0.45}$ | $95.51_{\pm0.45}$ | $95.51_{\pm0.45}$ |
| Re-EDL | $81.95_{\pm0.69}$ | $96.70_{\pm0.26}$ | $96.68_{\pm0.27}$ | $96.57_{\pm0.29}$ | $96.55_{\pm0.30}$ | $77.53_{\pm1.02}$ | $95.19_{\pm0.48}$ | $95.17_{\pm0.49}$ | $95.03_{\pm0.51}$ | $95.01_{\pm0.51}$ |
| Our | $\mathbf{84.25_{\pm0.25}}$ | $\mathbf{97.50_{\pm0.13}}$ | $97.38_{\pm0.12}$ | $97.02_{\pm0.17}$ | $92.32_{\pm0.63}$ | $\mathbf{79.04_{\pm0.50}}$ | $\mathbf{96.03_{\pm0.27}}$ | $95.88_{\pm0.27}$ | $95.44_{\pm0.30}$ | $89.67_{\pm0.58}$ |

| Method | CIFAR-10-C Level 4 | | | | | CIFAR-10-C Level 5 | | | | |
| --- | --- | --- | --- | --- | --- | --- | --- | --- | --- | --- |
| | Acc | MPU | DE | MI | VoE | Acc | MPU | DE | MI | VoE |
| EDL | $70.34_{\pm0.84}$ | $93.03_{\pm0.57}$ | $93.03_{\pm0.57}$ | $93.01_{\pm0.57}$ | $93.01_{\pm0.57}$ | $62.07_{\pm1.14}$ | $88.96_{\pm1.16}$ | $88.96_{\pm1.16}$ | $88.92_{\pm1.16}$ | $88.92_{\pm1.16}$ |
| $\mathcal{I}$-EDL | $69.07_{\pm1.19}$ | $92.30_{\pm0.74}$ | $92.30_{\pm0.74}$ | $92.23_{\pm0.74}$ | $92.23_{\pm0.74}$ | $60.40_{\pm1.16}$ | $87.75_{\pm1.10}$ | $87.75_{\pm1.10}$ | $87.65_{\pm1.11}$ | $87.64_{\pm1.11}$ |
| R-EDL | $71.41_{\pm0.91}$ | $93.11_{\pm1.00}$ | $93.11_{\pm1.00}$ | $93.11_{\pm1.00}$ | $93.11_{\pm1.00}$ | $62.42_{\pm1.43}$ | $88.66_{\pm1.94}$ | $88.66_{\pm1.94}$ | $88.66_{\pm1.94}$ | $88.66_{\pm1.94}$ |
| Re-EDL | $71.84_{\pm1.45}$ | $92.94_{\pm0.83}$ | $92.92_{\pm0.85}$ | $92.77_{\pm0.88}$ | $92.74_{\pm0.88}$ | $63.35_{\pm1.92}$ | $89.03_{\pm1.42}$ | $89.01_{\pm1.44}$ | $88.84_{\pm1.48}$ | $88.80_{\pm1.49}$ |
| Our | $\mathbf{72.88_{\pm0.91}}$ | $\mathbf{93.89_{\pm0.52}}$ | $93.74_{\pm0.57}$ | $93.20_{\pm0.61}$ | $86.13_{\pm0.88}$ | $\mathbf{63.89_{\pm1.19}}$ | $\mathbf{89.62_{\pm1.10}}$ | $89.52_{\pm1.17}$ | $88.78_{\pm1.19}$ | $79.65_{\pm1.59}$ |

# D  ADDITIONAL EXPERIMENTS

**UCE + MPU Achieves More Robust Performance Under Noisy Settings.** Following prior work Charpentier et al. (2020); Chen et al. (2023b); Deng et al. (2023), we evaluate our model under noisy data conditions. In particular, we follow Charpentier et al. (2020) and adopt the CIFAR-10-C benchmark Hendrycks & Dietterich (2019), which contains 15 types of image corruptions (e.g., Gaussian noise, blur, brightness) each applied at 5 severity levels. Table 4 reports the classification accuracy and misclassification detection performance on CIFAR-10 and CIFAR-10-C across all severity levels. As the corruption severity increases, classification accuracy inevitably degrades, making reliable misclassification detection especially critical. We observe that our method consistently surpasses previous EDL-based approaches across all corruption levels, both in terms of classification accuracy and misclassification detection. Notably, our proposed MPU achieves the strongest misclassification detection performance among all competing methods.

**Video-modality Verification.** To evaluate our method in a more challenging setting, we follow Bao et al. (2021) and perform experiments on the video-based open-set action recognition task, treating UCF-101 (Soomro et al., 2012) as known classes and HMDB-51 (Kuehne et al., 2011) as unknown. We report the Open maF1 and Open Set AUC metrics as defined in Bao et al. (2021), and all baseline results are taken from the same source. As shown in Table 5, our method (w/ MPU) achieves higher Open maF1 and AUC scores compared to prior approaches. In Fig. 5, we plot the average Open maF1 scores under varying openness by gradually introducing HMDB-51 test samples as unknowns. The results show that our MPU-based method consistently outperforms all baselines, and the performance gap remains large even as the openness level increases.

# E  DECISION BOUNDARY VISUALIZATION

We further compare the decision boundary learned by EDL with the UCE loss to those of C&S SVMs on three synthetic datasets from `sklearn`, namely two moons, concentric circles, and a linearly separable dataset. As shown in Fig. 9, we observe that for mostly linearly separable datasets, EDL exhibits strong alignment with the SVM. In contrast, for the nonlinearly separable concentric circles dataset, a notable discrepancy is observed, reflecting the distinct behavior of EDL in capturing nonlinear decision boundaries. Moreover, the classification accuracy achieved by EDL with the UCE loss is comparable to that of the C&S SVM across all three datasets.

Table 5: Performance of the video-modality setting using the I3D backbone on UCF-101, evaluated with HMDB-51 as unkown.

| Method | UCF-101 → HMDB-51 | |
| | Open maF1 | AUC |
| --- | --- | --- |
| OpenMax | 67.85±0.12 | 74.34 |
| MC Dropout | 71.13±0.15 | 75.07 |
| BNN SVI | 71.57±0.17 | 74.66 |
| SoftMax | 73.19±0.17 | 75.68 |
| RPL | 71.48±0.15 | 75.20 |
| DEAR | 77.24±0.18 | 77.08 |
| Our (w/ VoE) | 76.32±0.16 | 76.09 |
| Our (w/ MI) | 76.97±0.18 | 76.86 |
| Our (w/ DE) | 77.23±0.17 | 77.07 |
| Our (w/ MPU) | **78.31±0.18** | **77.67** |

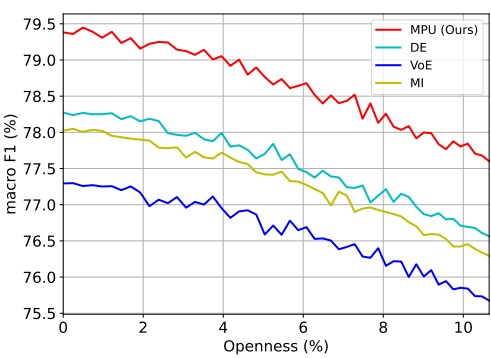

Figure 5: Open macro-F1 scores against varying Openness with different uncertainty measure when HMDB-51 as OOD dataset.

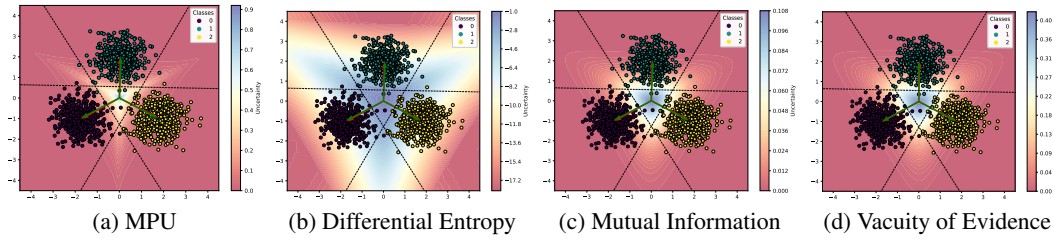

| (a) MPU | (b) Differential Entropy | (c) Mutual Information | (d) Vacuity of Evidence |

Figure 6: Comparison of different uncertainty measures on a toy dataset trained with the UCE loss. For visual consistency across panels, all uncertainty measures, including the normalized MPU, use the same `RdYlBu` colormap (red = low uncertainty, blue = high uncertainty). MPU assigns the highest uncertainty at the global center, where samples do not belong to any class, making it well-suited for out-of-distribution detection. It also yields elevated uncertainty near class decision boundaries, which is beneficial for misclassification detection. Differential Entropy exhibits a similar trend, showing high uncertainty at the global center; however, it also produces undesirably high uncertainty within in-distribution regions. Mutual Information and Vacuity of Evidence behave similarly, showing high uncertainty only near the global center. This is expected, as both reflect epistemic uncertainty, which becomes large when evidence for all classes is low.

## F    FUTURE WORK

Several promising directions emerge from our optimization-centric view of Evidential Deep Learning (EDL). First, while our analysis focuses on classification under stationary distributions, extending the optimization-consistency principle to non-stationary environments is an important avenue. In practical deployments, models frequently encounter distribution shifts and concept drift (Gama et al., 2014; LEARNING, 2009). Understanding how evidential objectives interact with drift adaptation and continual learning mechanisms (Yang et al., 2025b;a) may provide deeper insight into the stability of predictive uncertainty. Second, our proposed Margin-aware Predictive Uncertainty (MPU) explicitly models class-wise evidence separation. Future work could explore tighter connections between evidential uncertainty and representation geometry, building on studies of feature norm dynamics, angular margins, and hyperspherical embeddings (Liu et al., 2017; Wang et al., 2018; Yang et al., 2023). Such perspectives may clarify how margin formation and evidence strength jointly govern predictive confidence. Third, recent advances in representation learning highlight the importance of view construction, invariance, and robustness to partial observations (Chen et al., 2020; He et al., 2020; Yang et al., 2025c). Integrating these ideas with evidential training objectives may yield more stable and interpretable uncertainty behavior. Finally, our theoretical link between UCE minimization and maximum-margin solutions suggests broader opportunities at the intersection of margin theory, calibration, and rejection learning (Bartlett et al., 2006; Geifman & El-Yaniv, 2017).

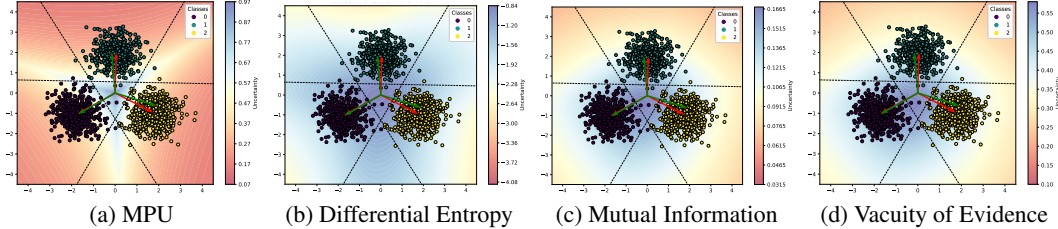

| (a) MPU | (b) Differential Entropy | (c) Mutual Information | (d) Vacuity of Evidence |

Figure 7: Uncertainty measures on a toy dataset trained with the Brier score loss. First, we observe that the classifier vectors learned with the Brier score (green arrows) are not tightly aligned with the C&S SVM solutions (red arrows), in contrast to the alignment observed under the UCE loss. This suggests that the implicit connection between EDL models and C&S SVM classifier geometry appears specifically under the UCE loss, rather than being a general property of all losses. In addition, MPU still exhibits its characteristic behavior: high uncertainty at the global center, followed by elevated uncertainty along class decision boundaries. This indicates that MPU remains applicable to models trained with the Brier score, although its effectiveness is weaker than under the UCE loss, due to the absence of the margin-aware structure that UCE naturally induces.



Figure 8: Predicted class probabilities of a logistic regression model on a toy dataset. The leftmost plot shows the final classification and confidence (max probability), while the three plots on the right show the individual probability surfaces for each class (0, 1, and 2).

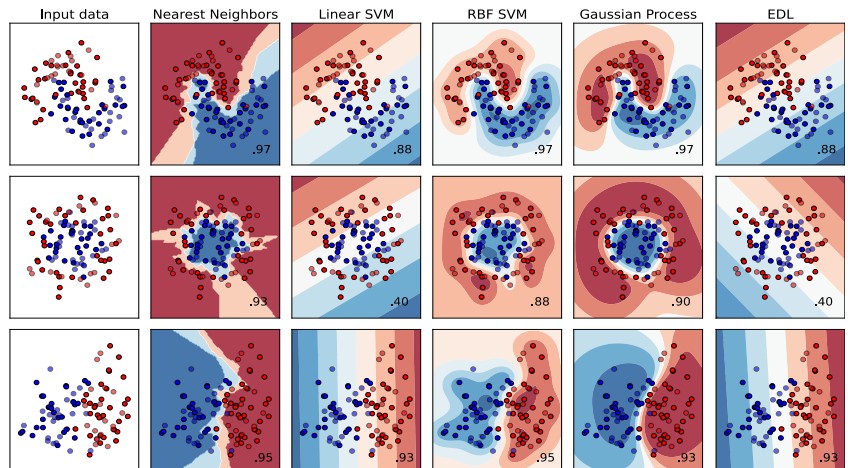

Figure 9: Comparison of the decision boundaries of several classifiers, on three synthetic datasets (two moons, concentric circles, and a linearly separable dataset), showing the distinct characteristics of each model's boundary.

Developing unified frameworks that jointly optimize margin, calibration, and abstention remains an exciting direction.

