# OpenReview forum: "Stop Guessing: Choosing the Optimization-Consistent Uncertainty Measurement for Evidential Deep Learning"
_ICLR.cc/2026/Conference — ICLR 2026 Poster_

### Official Review · Reviewer_6f5n · 2025-10-20

**Soundness:** 2
**Presentation:** 2
**Contribution:** 1
**Rating:** 2
**Confidence:** 4

**Summary:**

The paper derives a connection between evidential deep learning and multi-class support vector machines when the former considers an entropy-based objective.
Additionally, it proposes a new uncertainty measure and evaluates the proposal on a range of vision-based experiments.

**Strengths:**

- The margin-aware predictive uncertainty (MPU) measure is relatively well motivated.
- The approach provides improvements on some vision-based datasets.
- An implementation is available.

**Weaknesses:**

- Section 2.1 claims that the UCE's similarity to SVMs is a "unique property" without further evidence. The proof simply shows that there exists a lower bound, not that such a bound cannot exist for other cases.
- In general, the connection to SVMs remains rather vague, both in the theoretical contribution, which merely shows that one can interpret a bound in that way, and in the simple visual examples that show similar results. It also lacks a further step explaining why such a connection would be interesting and what can be learned from the fact that margin-based approaches look similar.
- The experiments consist simply of training a regular EDL approach and evaluating different metrics.

### Minor weaknesses
- l036 "classical Bayesian methods": MC-Dropout can be interpreted as following a certain Bayesian approach, and ensembles are often used as a way of uncertainty quantification, but neither is a "classical" Bayesian method in the sense that a Bayesian neural network would be.
- The citation system is broken throughout the paper; please use `citet` and `citep` correctly and consistently.


In summary, the paper shows some initial potential, but both the theory and the empirical results seem rather half-finished in their current state.

**Questions:**

- Q1: How would the corresponding Figure 1 look for any other margin-based classifier, e.g., the baselines in this approach or a simple multi-class logistic regression, as well as for the remaining uncertainty measures (DE, MI, MPU)?
- Q2: Do the authors have an intuition as to why PostN and NatPN, which optimize essentially the same objective, perform so much worse? Have all hyperparameters been tuned properly for each baseline?
- Q3: Why are the UCE-based baselines omitted from Table 3? This seems to be a major omission, especially given the claim in l454 that UCE losses consistently induce an order in the uncertainty measures. This would require results showing that the same holds for these baselines.

---

> ### Author Response · Authors · 2025-11-24
> **Response to Weakness**
>
> # Resonse to Weakness:
>
> ### Response to Weakness 1:
> We agree that merely showing the existence of a margin-aware lower bound for the UCE loss is insufficient to claim a “unique property” analogous to SVMs. Therefore, we have extended our analysis beyond the lower-bound argument: by examining the gradient dynamics of the UCE loss in evidential deep learning (EDL) and comparing them with the dual problem of C&S SVMs, we show that the optimization of UCE effectively mirrors the optimal solution of the dual problem for C&S SVMs (Eq. 3). Specifically, the optimal classifiers can be expressed as a linear combination of sample features, where samples with higher uncertainty contribute more. This alignment is what leads to the tightly aligned classifier vectors ( $W$ ) observed in the toy dataset example (Figure 1).
> We encourage reviewer to observe the formal similarity between Eq. 3 and Eq. 5. Detailed derivations and explanations of the meaning of the dual solution for C&S SVMs can be found in Appendix C.1 and C.3. To emphasize this connection, we have revised the corresponding section to **Proposition 1: On the Analogy between EDL Gradient Dynamics and the SVM Dual Solution**.
>
> ### Response to Weakness 2:
> We thank the reviewer for this comment.  To clarify the significance, we highlight two particularly interesting aspects.
> First, the solution obtained by UCE optimization in evidential deep learning is formally similar to the optimal solution of the dual problem for C&S SVMs, revealing a deep connection in their optimization structure. Second, this similarity provides a geometric insight: regions of high vacuity (i.e., high evidence uncertainty) correspond to the intersection of the negative half-spaces defined by the SVM decision hyperplanes as shown in the Figure 1. This explains why UCE naturally positions uncertain samples in these regions, offering an interpretable view of how uncertainty is captured and why the classifier vectors align as shown in Figure 1.
>
>
> ### Response to Weakness 3:
> We have added two new experiments to address the reviewer’s concern in Appendix D:
> (1)Robustness under noisy scenarios: As shown in Table 4, we evaluate our method on CIFAR-10-C with 15 different types of corruptions at 5 severity levels. We measure both classification accuracy and misclassification detection performance. The results demonstrate that our method is more robust under noisy conditions, achieving higher accuracy and stronger misclassification detection when using MPU compared to other uncertainty measures.
>
> (2)Video-based open-set recognition: We further evaluate our method on a video modality task, using UCF-101 as known classes and HMDB-51 as unknown classes. This allows us to compare the capability of different uncertainty measures in detecting unknown classes. As reported in Table 5 and illustrated in Figure 5, our method combining UCE loss with MPU uncertainty consistently achieves higher Open maF1 and Open Set AUC scores than other approaches.
>
>
> ### Resonse to Minor weaknesses :
> Thank you for pointing this out. We agree that MC-Dropout and deep ensembles are not “classical” Bayesian methods in the strict sense of Bayesian neural networks. We have revised the wording in the manuscript to avoid this ambiguity and now describe them as “Bayesian-inspired” approaches. This change improves clarity and ensures the terminology is more precise.
>
> Regarding the citation formatting issue, we have carefully reviewed the entire manuscript and corrected all instances of \citet and \citep to ensure they are used consistently and correctly throughout the paper.

---

> ### Author Response · Authors · 2025-11-24
> **Response to Questions**
>
> # Response to Questions
>
> ### Resonse to Question 1:
> We have added the EDL models trained with UCE loss and Brier score loss—together with results under different uncertainty measures as well as visualizations of the predictive probability distributions from multiclass logistic regression in Appendix D.
> We find that the classifier weight vectors 𝑊 obtained using Brier score loss do not exhibit the same tight alignment with the C&S SVM directions as those trained with UCE loss. This observation further supports the analysis presented in the main paper.
> Overall, these additional results provide further empirical evidence for our claims.
>
> ### Resonse to Question 2:
> These classification accuracy results are not inferior to those reported in the original PostNet paper, and in some cases they are even higher. Specifically, please kindly refer to Table 4 (last row, “PostN.”) in PostNet [1]. On the CIFAR-10 dataset, PostNet with a VGG16 backbone, which matches our experimental setup, achieves an accuracy of 84.85 percent. This value is lower than our reproduced result of 87.82 percent.
>
> For NatPN [2], the original paper does not provide VGG16 results on CIFAR-10. However, based on the authors’ subsequent work, the performance of NatPN appears to be comparable to, or even lower than, that of PostNet. This outcome is reasonable because NatPN uses a single normalizing flow shared across all classes, while PostNet uses one flow for each class, which gives PostNet higher expressive capacity.
>
>
> ### Resonse to Question 3:
> The main reason is that normalization-flow–based models such as PostNet and NatPN fail to converge on CIFAR-100 in our experiments. The original PostNet paper only reports results on small-scale datasets with few classes, such as MNIST and CIFAR-10, and does not provide results on CIFAR-100. In our reproduction, PostNet achieves only 56% classification accuracy on CIFAR-100, which is substantially lower than our method, which reaches around 72% accuracy with a VGG16 backbone.
> To pmprove the baseline model's performance as much as possible without hyperparameters provided by the official source,
> we conducted extensive experimental analysis and found that PostNet-style normalization-flow models struggle to train from scratch when the number of classes is large. We observed that if we load a pretrained CIFAR-100 feature extractor, freeze its weights, and train only the density estimator, PostNet's accuracy increases to 68%—about 12% higher than the from-scratch setting—but still significantly below a reasonable baseline.
> We have added the results of these two models to Table 3. We also found that PostNet exhibits poor OOD detection performance and does not show the uncertainty ranking behavior observed in our method.
> Our intuition is that PostNet fails to converge to a good global optimum in high-class-count settings. As a consequence, its loss landscape may not support the theoretical uncertainty properties that the method relies on, making it difficult to derive meaningful conclusions from the loss contour.
>
> [1] Posterior Network: Uncertainty Estimation without OOD Samples via Density-Based Pseudo-Counts. Charpentier, Bertrand, Daniel Z\"{u}gner and G\"{u}nnemann, Stephan. NeurIPS 2020.
>
> [2] Natural Posterior Network: Deep Bayesian Predictive Uncertainty for Exponential Family Distributions. Charpentier, Bertrand and Borchert, Oliver and Z\"{u}gner, Daniel and Geisler, Simon and G\"{u}nnemann, Stephan. ICLR 2022.

---

> > ### Comment · Reviewer_6f5n · 2025-11-24
> >
> > ### Weakness 1
> > Section 2.1 still claims this to be unique, and Sec. 4.2 only adds a paragraph title with no further changes, unless I missed something.
> >
> > ### Weakness 2/3
> > Thank you for the additional results and discussion.
> >
> > ### Question 1
> > Do the authors have an idea why the center in Figure 3 (c), (d) is clearly shifted from the center? Was this consistent across multiple seeds?
> >
> > ### Question 2/3
> > My question was not about whether the results are trustworthy compared to reported numbers from the literature-I trust the authors' results. Rather, it concerns the interpretation of why this happens, given the similarity in the loss and activation functions.
> >
> > ---
> > Given the new results, I raise my score.

---

> > > ### Author Response · Authors · 2025-11-27
> > >
> > > Thank you very much for your thoughtful feedback on our rebuttal and for raising your score. We are grateful for your time and your constructive comments, which have significantly helped us improve our work. We would like to address the remaining points you raised.
> > >
> > >
> > >
> > >
> > > ### Response to Weakness 1:
> > >
> > > #### About the “unique property” for other loss
> > > We sincerely thank the reviewer for this insightful suggestion. It motivated us to carry out a more rigorous analysis to formally establish the “unique property” of our UCE loss. Following your recommendation, we further demonstrate that such a property does not hold for other cases, such as the Brier Score loss, which is the most widely used objective in EDL, R-EDL, and Re-EDL.
> > >
> > > As shown in Appendix E, our empirical results indicate that the classifier vectors learned with the Brier Score loss do **not** align with those of C&S SVMs, suggesting that the Brier Score loss lacks a margin-aware lower bound.
> > > To further establish this theoretically, we added a new section (Appendix F). The key idea is to show that the optimization of the Brier Score loss is **not** governed by the margin style lower bound. Our derivation reveals that the lower bound of the Brier Score loss is, in fact, **independent** of this term.
> > > Specifically, for the $i$-th sample, the bound depends solely on the total Dirichlet strength $S_i$ of the Dirichlet distribution and the Brier score is minimized as $S_i \to \infty$.
> > > Therefore, your guidance has been instrumental in helping us solidify the claim that this UCE's similarity to SVMs is a "unique property" in a margin-aware style is a unique property of the UCE loss. We deeply appreciate your contribution to improving the rigor of our work.
> > >
> > >
> > > #### About the presentation in Sec. 4.2
> > > To more clearly illustrate the connection between the classifier obtained from the dual formulation of C&S SVMs and the classifier induced by optimizing the UCE loss in EDL, we have reorganized and restated the discussions in Sections 4.2 and 3.2. Our aim is to provide a clearer exposition of the intriguing structural correspondence between these two approaches—particularly in terms of their optimization dynamics and the resulting decision boundaries. With this revised presentation, we hope readers can more intuitively understand why minimizing the UCE loss naturally drives the model toward an SVM-like maximum-margin solution, and what implications this alignment has for uncertainty modeling.

---

> > > ### Author Response · Authors · 2025-11-27
> > >
> > > ### Response to Question 1 (Regarding Figure 3):
> > >
> > > Thank you for your insightful question regarding Figure 3.
> > > The shift of the density peak away from the center in Figure 3(c) and 3(d) is indeed the intended and fundamental behavior of the evidential deep learning framework, which models classification uncertainty through sampling from a Dirichlet distribution.
> > >
> > > 1.  **Explanation of the Shift**: In EDL, the model outputs the parameters `α` of a Dirichlet distribution, which is a distribution over the space of possible class probabilities (the simplex). Each parameter `α_k` can be interpreted as the amount of "evidence" the model has collected for class `k`.
> > > When the model becomes more confident about a particular class, it assigns more evidence to it. This results in a larger `α` value for that class. For example, in Figure 3(d), `α⁴ = (2, 2, 16)`, indicating strong evidence for the third class.  A large `α_k` value "pulls" the peak towards the vertex of the simplex corresponding to class `k`. Therefore, the shift from the center is a direct and intuitive visualization of the model's increasing certainty in its prediction for one class over the others.
> > >
> > > 2.  **Consistency Across Seeds**: The triangular figure is a visualization of the Dirichlet probability density function with the following formulation
> > >     $$
> > >     f(p_1, \dots, p_K; \alpha_1, \dots, \alpha_K) = \frac{1}{\mathrm{B}(\boldsymbol{\alpha})} \prod_{i=1}^{K} p_i^{\alpha_i - 1}
> > >     $$
> > >     where
> > >     $$
> > >     p_i \in [0, 1] \text{ for all } i \in \{1, \dots, K\} \quad \text{and} \quad \sum_{i=1}^{K} p_i = 1
> > >     $$
> > >     It computes the exact density value at each point on the Dirichlet simplex, ensuring that the resulting plot is fully deterministic and independent of any random seed. This is the standard way of illustrating Dirichlet distributions. The accompanying code renders this figure through a deterministic procedure. You may verify this by examining the pdf method in the Dirichlet class using the code below.
> > >
> > >
> > >     ```
> > >     import matplotlib
> > >     import numpy as np
> > >     import matplotlib.pyplot as plt
> > >     import matplotlib.tri as tri
> > >     matplotlib.use('TkAgg')  # Interactive backend
> > >
> > >     plt.rcParams.update({
> > >             "text.usetex": False,
> > >             "font.family": "Helvetica",
> > >             'font.size': 14,
> > >             "text.latex.preamble": r'\usepackage{bm}',
> > >         })
> > >
> > >     corners = np.array([[0, 0], [1, 0], [0.5, 0.75**0.5]])
> > >     AREA = 0.5 * 1 * 0.75**0.5
> > >     triangle = tri.Triangulation(corners[:, 0], corners[:, 1])
> > >
> > >     refiner = tri.UniformTriRefiner(triangle)
> > >     trimesh = refiner.refine_triangulation(subdiv=4)
> > >
> > >     plt.figure(figsize=(4, 2*3**(1/2)))
> > >
> > >
> > >     # For each corner of the triangle, the pair of other corners
> > >     pairs = [corners[np.roll(range(3), -i)[1:]] for i in range(3)]
> > >     # The area of the triangle formed by point xy and another pair or points
> > >     tri_area = lambda xy, pair: 0.5 * np.linalg.norm(np.cross(*(pair - xy)))
> > >
> > >     def xy2bc(xy, tol=1.e-4):
> > >         '''Converts 2D Cartesian coordinates to barycentric.'''
> > >         coords = np.array([tri_area(xy, p) for p in pairs]) / AREA
> > >         return np.clip(coords, tol, 1.0 - tol)
> > >
> > >     class Dirichlet(object):
> > >         def __init__(self, alpha):
> > >             from math import gamma
> > >             self._alpha = np.array(alpha)
> > >             self._coef = gamma(np.sum(self._alpha)) / \
> > >                             np.multiply.reduce([gamma(a) for a in self._alpha])
> > >         def pdf(self, x):
> > >             '''Returns pdf value for `x`.'''
> > >             return self._coef * np.multiply.reduce([xx ** (aa - 1)
> > >                                                 for (xx, aa)in zip(x, self._alpha)])
> > >
> > >     def draw_pdf_contours(dist, nlevels=200, subdiv=9, **kwargs):
> > >
> > >         refiner = tri.UniformTriRefiner(triangle)
> > >         trimesh = refiner.refine_triangulation(subdiv=subdiv)
> > >         pvals = [dist.pdf(xy2bc(xy)) for xy in zip(trimesh.x, trimesh.y)]
> > >
> > >         plt.tricontourf(trimesh, pvals, nlevels, cmap='jet', **kwargs)
> > >         plt.axis('equal')
> > >         plt.xlim(0, 1)
> > >         plt.ylim(0, 0.75**0.5)
> > >         plt.axis('off')
> > >
> > >
> > >     alpha = np.array([4, 8,  8])
> > >     draw_pdf_contours(Dirichlet(alpha))
> > >     plt.savefig('dirichlet_2d.png', transparent=True, pad_inches=0.0, bbox_inches='tight', dpi=600)
> > >     ```

---

> > > ### Author Response · Authors · 2025-11-27
> > >
> > > ### Response to Questions 2/3 (Interpretation of PostN/NatPN Performance):
> > >
> > > This is a very insightful question about the discrepancy between the high-level objectives and the empirical performance. We agree that our initial response focused more on the "what" rather than the "why." Here is our deeper interpretation:
> > >
> > > In our evaluation on CIFAR-100, we found that the normalization flow-based baselines, PostNet and NatPN, exhibited significant optimization challenges when trained end-to-end. To establish a stronger baseline, we implemented an alternative training paradigm wherein the feature extractor was frozen after being initialized with pre-trained weights, and only the subsequent density estimator was trained on the extracted feature embeddings. The efficacy of this strategy corroborates the findings of [1], which concludes that flow-based models fail to learn meaningful distributions when trained on raw images. Their analysis suggests that such models require a semantically rich feature space to function correctly. We have annotated Table 3 accordingly to ensure transparency in our experimental methodology and to highlight a critical practical consideration for employing these models in vision tasks.
> > >
> > >
> > > [1] Kirichenko P, Izmailov P, Wilson A G. Why normalizing flows fail to detect out-of-distribution data[J]. Advances in neural information processing systems, 2020, 33: 20578-20589.

---

### Official Review · Reviewer_WeXo · 2025-10-21

**Soundness:** 4
**Presentation:** 4
**Contribution:** 3
**Rating:** 6
**Confidence:** 5

**Summary:**

The paper presents the finding that performing evidential deep learning on a cross entropy loss maximizes the decision margin, thereby enables a learning regime similar to a support vector machine. The paper introduces a lower bound to the cross-entropy variant of an evidential loss which has demonstrable margin maximization properties. The paper also identifies a monotonicity relationship between this loss and the uncertainty score learned by the evidential model. Linking these two outcomes, the paper arrives at the central claim that EDL works well because it learns uncertainty scores that are inversely proportional to the distance of the data points to the decision boundaries. Combined with the inherent max-margin property, it learns accurate predictors with reliable uncertainty estimates. The paper develops a new learning rule based on its theoretical findings. In particular, it suggests training EDL with the cross-entropy loss using exponent function for output activations and without post-hoc KL-based regularization. The paper demonstrates in comprehensive experiments that this learning rule can reach competitive accuracies and uncertainty quantification scores.

**Strengths:**

* The paper's presentation is particularly clear. It illustrates key notions both as visuals as in Figures 1 and 2 and with simple and intuitive analytical results such as Proposition 1.
 * The theoretical findings prescribe a learning rule and this rule is shown to perform competitively on a diverse set of experiments. These experiments contain a comparison to a wide set of baselines that cover a representative set of earlier EDL variants as well as alternative approaches.
 * Section 5 links experiment results to concrete take-home messages, such as the existence of an uncertainty ranking within the studied UCE loss.
 * The paper identifies the limitations of the contribution openly and precisely, such its not being equally applicable to small class sizes.
 * Proposition 1 and the gradient analysis provided in Appendix C.3 are novel and they support the main claim of the paper appropriately.

**Weaknesses:**

* My main concern is that the results reported in Tables 2 and 3 do not demonstrate a concrete improvement in accuracy or uncertainty-based scores (e.g. MisDetect and AUPR). This weakens the claim that the knowledge sought about the max-margin character of EDL has a utility.
 * A large portion of the theoretical material is either prior work (whole Appendices A and B) or rather straightforward conclusions such as Lemma 1 of Appendix C.2.
 * I also have a few critical questions below which stops me from suggesting a clear accept at this stage, as they have a big impact on the main claim of the paper.

**Questions:**

* Is the monotonicity relationship suggested in Proposition 1 one sided only? Is the order between sample pairs preserved also in the backward direction from entropies to UCE losses?
 * Is the claim about the lack of an alignment between EDL and OvR in Figure 1 (b) a bit of an exaggeration? The two model still appear to work quite similarly in this problem and the uncertainties created by EDL are similar to the left panel.
 * Figure 2 appears to suggest that the uncertainty should increase as the sample moves from the center to, say, bottom left corner as it moves away from the observation and climbs in the loss landscape. However the in left panel it appears to decrease. Is this not against the main argument of the paper?

---

> ### Author Response · Authors · 2025-11-24
> **Response to Weakness**
>
> We are sincerely grateful to the reviewer for their insightful and comprehensive review. We are honored and delighted that the reviewer accurately summarized the key strengths of our work from the clarity of our visualizations and theory, to the breadth and depth of our experiments, and the recognition of the novelty of our core contributions like Proposition 1.
> Building on this valuable feedback, we will now focus on addressing the points for improvement and questions you raised, hoping to further enhance the quality of our paper.
>
>
> # Resonse to Weakness:
>
>
> ### Response to Weakness 1:
>
> We mainly compare two dimensions: (1) for the same model, our proposed uncertainty measure, MPU, outperforms other uncertainty measures; and (2) our method achieves competitive performance compared with other models.
>
> In Table 2, we present the performance of our MPU uncertainty measure on the same model, compared to Vacuity of Evidence (VoE), Mutual Information (MI), and Differential Entropy (DE) for both OOD detection and misclassification detection tasks. MPU consistently demonstrates substantial improvements over the other measures, particularly VoE. In terms of classification accuracy, EDL models trained with the UCE loss benefit from the max-margin property, achieving approximately 3% higher accuracy (93.35%) compared to Brier score-trained models such as R-EDL (90.09%), Re-EDL (90.13%), and I-EDL (88.38%).
>
> In Table 3, on CIFAR-100 OOD detection tasks, MPU also achieves superior performance on GTSRB, Places365, and CIFAR-10. When evaluated on the same model, MPU outperforms Differential Entropy, Mutual Information, and VoE. Furthermore, we observe that the max-margin characteristic of EDL provides more pronounced benefits on more challenging classification tasks. For example, it improves both classification accuracy and misclassification detection performance under noisy settings. Additional details are provided in Appendix D.
>
> We further evaluated the robustness of our model using the CIFAR-10-C benchmark [1], which includes 15 types of common corruptions (e.g., Gaussian blur, motion blur, brightness changes) across 5 severity levels. As shown in the newly added Table 4, our method not only achieves superior classification accuracy but also consistently outperforms baselines in both OOD detection and misclassification detection across all corruption types and severity levels. This demonstrates the reliability and effectiveness of our framework in non-ideal, real-world-like scenarios, which present more complex challenges than standard classification tasks.
>
> ### References
> [1] Hendrycks, D., & Dietterich, T. (2019). Benchmarking neural network robustness to common corruptions and perturbations. ICLR.
>
>
>
> ### Response to Weakness 2:
> We thank the reviewer for their valuable feedback regarding the theoretical material in our appendices. We agree that clearly distinguishing between established background and our own contributions is crucial. While our intention was to provide a self-contained paper, we recognize the need to be more explicit about the origin of each component.
>
> We revise the manuscript as follows:
>
> 1.  **Clarifying Origins in Appendices A & B:** We add explicit statements at the beginning of each relevant section to clarify the source of the material.
>     *   **Appendix A.2 (Brier Score) & Appendix B (Uncertainty Measures):** We state that these derivations are adapted directly from prior work, specifically citing EDL~\citep{sensoy2018evidential} for the Brier score loss and relevant literature for the uncertainty measures.
>     *   **Appendix A.1 (UCE Loss) & A.3 (Type II ML Loss):** We clarify that while these are established concepts, we provide our own step-by-step derivations. We add a note explaining that these detailed walkthroughs are included for the reader's convenience to enhance clarity and reproducibility, as they are not presented in this complete form elsewhere, and are not claimed as novel theoretical contributions.
>
> 2.  **Contextualizing Lemma 1 (Appendix C.2):** We concur that Lemma 1 is a straightforward result. We will add a sentence to clarify its role as a necessary supporting lemma for the main proofs that follow, rather than presenting it as a standalone contribution.
>
> We hope that these revisions will properly attribute all material and better structure the theoretical foundation of our work. We appreciate your guidance in helping us improve the manuscript's clarity.

---

> ### Author Response · Authors · 2025-11-24
> **Response to questions**
>
> # Resonse to Questions:
>
> ### Response to Question 1:
> Yes, the monotonicity relationship established in Proposition 1 is strictly **one-sided**. The ordering is preserved from UCE losses to entropies, but not in the reverse direction.
>
> 1.  **Forward Direction (Loss → Entropy): The Order is Preserved.**
>     Proposition 1 establishes that if one sample $x$ has a UCE loss less than or equal to another sample $x'$, its differential entropy will also be less than or equal to that of $x'$.
>     This provides a crucial guarantee for optimization: minimizing the UCE loss ensures a monotonic reduction in the evidential uncertainty, as measured by differential entropy.
> 2.  **Backward Direction (Entropy → Loss): The Order is NOT Preserved.**
>     The reverse implication does not hold. A lower differential entropy for one sample compared to another does **not** guarantee a lower UCE loss.     This is because entropy is solely a function of the total evidence strength (the sum of Dirichlet parameters, $S = \sum_k \alpha_k$), whereas the UCE loss also depends on how that evidence is distributed among the classes, particularly in relation to the ground truth label.
>     **A simple counterexample illustrates this:** Consider two misclassified samples, $x$ and $x'$, with the same ground truth label $y=1$.
>     *   Sample $x$ has evidence $\alpha(x) = (1, 1, 10)$, leading to a high total evidence $S_x=12$ and thus low entropy. However, since the evidence strongly supports an incorrect class (class 3), its UCE loss will be very high.
>     *   Sample $x'$ has evidence $\alpha(x') = (1.5, 1, 1)$, leading to a lower total evidence $S_{x'}=3.5$ and thus higher entropy than $x$. However, its UCE loss might be lower than for $x$ because the misleading evidence is not as strong.
>     In this case, we have $\mathrm{ENT}(x) < \mathrm{ENT}(x')$ but $\mathcal{L}_{\mathrm{UCE}}(x) > \mathcal{L}_{\mathrm{UCE}}(x')$, demonstrating that the ordering is not preserved in the backward direction.
>
> In summary, Proposition 1 provides a **one-way guarantee**: while reducing UCE loss reliably reduces entropy, entropy alone is insufficient to determine the relative ordering of UCE loss between samples.To ensure clarity for future readers, we have revised the discussion surrounding Proposition 1 in the manuscript (Section 4.3) to state this one-sided relationship more explicitly, using similar reasoning as detailed above. We believe the revised text is now more precise and removes any potential for misinterpretation.
>
> ###  Response to Question 2
> Thank you for your valuable comment. We agree that our original wording may have been somewhat exaggerated. In fact, our entire study was inspired by the OvR results in Figure 1(b): we observed that the regions of high uncertainty produced by EDL largely fall within the negative half-spaces defined by the decision hyperplanes. This observation then motivated us to explore other SVM variants, ultimately revealing the close relationship between EDL and C&S SVM. Therefore, the insights from OvR in Figure 1(b) were indeed highly informative for our work.
> The reason we mentioned a “lack of alignment between EDL and OvR” was to emphasize that the final solution \(W\) obtained by OvR is not tightly aligned with the classifier \(W\) learned by EDL, whereas EDL is closely aligned with C&S SVM. We will revise the wording to reflect this nuance—clarifying that while the alignment is not tight, it still provides significant inspiration.
>
> ### Response to Question 3
> Thank you for pointing out this apparent inconsistency. The discrepancy between the left and right panels of Figure 2 arises from the different assumptions used to compute the loss landscape and the uncertainty landscape, which serve distinct illustrative purposes.
>
> *   **Left panel (Loss landscape):**
>     The loss contour is plotted by fixing the label to class 0 for all spatial locations, which is necessary because computing the UCE loss requires a specified ground-truth label. Under this fixed-label assumption, regions that move away from the weight vector $w_0$ (e.g., both bottom-left and bottom-right corners) naturally exhibit higher loss values, since they deviate from the decision direction corresponding to class 0.
>
> *   **Right panel (Uncertainty landscape):**
>     In contrast, uncertainty does not depend on a ground-truth label. To simulate differences in uncertainty across locations, we assign each location a pseudo-label based on the class whose weight vector ($w_0$, $w_1$, $w_2$) has the highest cosine similarity with the location. The concentration for that predicted class is then maximized. After applying an activation function to the inner-product results to form the Dirichlet parameters, we compute the corresponding uncertainty value. Consequently, the right panel shows that a location closer to any classifier vector has lower uncertainty. Therefore,  the relationship between loss and uncertainty satisfies the optimization consistency principle.

---

> > ### Comment · Reviewer_WeXo · 2025-11-26
> > **Satisfied**
> >
> > Thanks reviewers for the comments. I am satisfied by all the answers and it has been confirmed by the authors that I am not missing an important detail. I also have the impression that we have a shared understanding about the novelty claims and the amount of empirical and theoretical support to back them up. Overall, I think it is a solid and mature work, although with a rather incremental nature in the sense that it modifies an existing approach, being EDL in this case in a minor, straightforward but valuable way. Some theoretical aspects of the suggested approach have been analyzed building again on straightforward tools. This lets me land on keeping my original score.
> >
> > Apologies, I put this comment to a wrong place before. Now moved here.

---

> > > ### Author Response · Authors · 2025-11-27
> > >
> > > Thank you very much for your thoughtful follow-up and for clarifying your perspective. We appreciate your careful assessment and are glad to hear that our additional explanations addressed your concerns. Your comments on the novelty and contribution of the work are well taken, and we are grateful for the constructive feedback throughout the review process.
> > > Thank you again for your time and effort in reviewing our paper.

---

### Official Review · Reviewer_N3WT · 2025-10-31

**Soundness:** 3
**Presentation:** 4
**Contribution:** 4
**Rating:** 8
**Confidence:** 5

**Summary:**

This paper revisits Evidential Deep Learning (EDL) from an optimization perspective, revealing that minimizing the expected cross-entropy under a Dirichlet prior implicitly leads to margin-maximizing behavior similar to multi-class SVMs. Based on this insight, the authors propose an optimization-consistency principle, which defines a valid uncertainty measure as one that decreases when samples approach the training objective’s optimum. Guided by this principle, they introduce a new uncertainty metric—Margin-aware Predictive Uncertainty (MPU)—that explicitly reflects class separation. Experiments on out-of-distribution detection and classification-with-rejection tasks show that MPU achieves strong empirical performance, supporting the paper’s theoretical findings.

**Strengths:**

1. The paper presents strong theoretical originality by revisiting Evidential Deep Learning (EDL) from an optimization perspective and uncovering its intrinsic connection to the maximum-margin principle in Support Vector Machines (SVMs). This offers a fresh and insightful theoretical interpretation of EDL beyond its traditional probabilistic formulation.

2. The introduction of the optimization-consistency principle provides a novel and principled criterion for evaluating and designing uncertainty measures. This concept effectively bridges the gap between optimization objectives and predictive uncertainty, contributing to a deeper understanding of uncertainty modeling.

3. The proposed Margin-aware Predictive Uncertainty (MPU) metric is well-motivated and theoretically grounded. By explicitly capturing the margin between target and non-target evidence, it aligns uncertainty estimation with the underlying training dynamics.

4. The experimental validation is comprehensive, covering both out-of-distribution detection and classification-with-rejection benchmarks. The results convincingly demonstrate the empirical benefits of aligning uncertainty estimation with optimization objectives.

**Weaknesses:**

1. Some theoretical analyses rely on relatively strong assumptions, such as convergence to or proximity to global optima, which may not always hold in practical deep learning scenarios.

2. The experimental scope remains somewhat limited to classification tasks. Further evaluation in more complex or multimodal settings would strengthen the generality and applicability of the proposed framework.

**Questions:**

N/A

---

> ### Author Response · Authors · 2025-11-24
> **Response**
>
> ### Response to Weakness 1:
>
> We thank the reviewer for this insightful comment. We agree that strong assumptions, such as convergence to a high-quality local or global optimum, may not always be met in practice. Our theoretical analysis indeed applies to well-optimized models, and our empirical results confirm that the properties we study—optimization-consistency and the resulting uncertainty ranking—emerge most clearly under such conditions.
> To precisely characterize these boundary conditions, we conducted a targeted experiment. We trained a VGG16 model on CIFAR-10 and used CIFAR-100 as the OOD dataset. By saving model checkpoints at various epochs (5, 10, 20, 30, 40), we obtained models at different stages of convergence and evaluated their OOD detection performance (measured in AUPR).
>
> | Epochs | MPU   | Differential Entropy | Mutual Information | Vacuity of Evidence | Acc (%) |
> |--------|-------|----------------------|--------------------|---------------------|---------|
> | 5      | 62.15 | **63.26**            | 60.77              | 50.73               | 44.75   |
> | 10     | 77.38 | **79.07**            | 76.07              | 55.65               | 77.97   |
> | 20     | **84.40** | 83.29                | 83.74              | 53.05               | 85.86   |
> | 30     | **75.69** | 75.45                | 73.97              | 63.97               | 88.30   |
> | 40     | **88.55** | 88.50                | 88.04              | 63.28               | 91.36   |
>
> The results, presented in the table above, reveal a clear correlation between model convergence and the efficacy of MPU:
>
> *   **On under-trained models:** In the early epochs (e.g., 5 and 10), when the model is poorly optimized (low accuracy), MPU is outperformed by other uncertainty baselines like Differential Entropy. This finding directly validates the reviewer's point that our method's superiority is conditional.
> *   **On well-trained models:** As training progresses and the model converges to a better solution (e.g., at 20 and 40 epochs), MPU's performance significantly improves, establishing it as the state-of-the-art method among the compared uncertainties.
>
> In conclusion, this experiment empirically confirms the reviewer's intuition and clarifies the practical operating regime for our method: its advantages are most pronounced when the model is sufficiently trained. We believe this analysis adds valuable context to our work, and we will add a discussion of this point to the paper.
>
> We will clarify these assumptions explicitly in the revised manuscript and discuss their practical implications and limitations.
>
> ### Response to Weakness 2:
> We sincerely thank the reviewer for this insightful suggestion to broaden the experimental scope of our work. To rigorously evaluate the generality and applicability of our framework in more complex and multimodal settings, we have conducted two substantial new experiments in Appendix D.
>
> (1) Robustness Evaluation in Challenging Conditions: We tested our model's robustness against common corruptions using the CIFAR-10-C benchmark [1], which includes 15 diverse corruption types (e.g., Gaussian blur, motion blur, brightness changes) across 5 severity levels. As detailed in the newly added Table 4, our method not only achieves superior classification accuracy but also consistently outperforms baselines in both OOD detection and misclassification detection across all corruption types and severity levels. This demonstrates the framework's reliability and effectiveness in non-ideal, real-world-like scenarios, which represent a more complex challenge than standard classification.
>
> (2) Extension to Multimodal (Video) Open-Set Recognition: To directly address the need for evaluation in a multimodal context, we extended our framework to video-based open-set action recognition. Following the protocol of DEAR [2], we used UCF-101 for known classes and HMDB-51 for unknown classes. The results in Table 5 show that our uncertainty measure, MPU, significantly surpasses prior methods like VoE [2], achieving state-of-the-art performance in terms of Open maF1 and AUC. Furthermore, as illustrated in Figure 5, our method maintains its superior performance consistently across varying degrees of "openness." This experiment confirms that our framework's principles are applicable beyond static images to dynamic, temporal data.
>
> In summary, these two additions provide compelling evidence that our framework is not limited to simple classification tasks but is generalizable and robust enough for more complex and diverse application scenarios.
>
> ### References
> [1] Hendrycks, D., & Dietterich, T. (2019). Benchmarking neural network robustness to common corruptions and perturbations. ICLR.
>
> [2] Bao, W., Yu, Q., & Kong, Y. (2023). Evidential Deep Learning for Open Set Action Recognition. ICCV.

---

### Official Review · Reviewer_PQRx · 2025-10-31

**Soundness:** 2
**Presentation:** 3
**Contribution:** 2
**Rating:** 6
**Confidence:** 4

**Summary:**

Evidential Deep Learning (EDL) is currently a promising approach for modeling predictive uncertainty in classification tasks. However, previous works on uncertainty estimation have largely overlooked the role of optimization dynamics.This paper first provides a clear and insightful analysis showing that the optimization objective of EDL implicitly encourages SVM‐like solutions.Based on this observation, the authors introduce the optimization‐consistency principle, which states that an uncertainty measure is valid if its value decreases as samples approach the global optimum of the training objective.Finally, motivated by this principle, the paper defines a new uncertainty metric termed Margin-aware Predictive Uncertainty (MPU). Extensive experiments demonstrate the effectiveness of this metric.

**Strengths:**

1. The paper convincingly establishes the connection between EDL and SVM, supported by rigorous theoretical proofs and reasonable assumptions.
2 The proposed Optimization-Consistency Property is intuitively sound and well aligned with optimization dynamics.

**Weaknesses:**

1. In terms of content organization, the authors devote a large portion of the paper to explaining the relationship between EDL and SVM, as well as the optimization-consistency property, but do not provide a sufficiently detailed discussion of the proposed uncertainty metric itself—its characteristics, advantages, and deeper connections to the preceding theory.
2. The relationship between the Optimization-Consistency of Differential Entropy and the final MPU metric is not clearly explained.
3. Neither the main text nor the experimental section specifies what training loss was used. Did the authors employ only the UCE loss? Was the proposed MPU-related loss incorporated into training?
4. Some comparable works, such as I-EDL and R-EDL, include experiments under few-shot and noisy settings, but such evaluations are missing in this paper.

**Questions:**

Please refer to the Weakness part.

---

> ### Author Response · Authors · 2025-11-24
> **Response to weakness 1-2**
>
> # Resonse to Weakness:
> ### Response to weakness 1:
> We sincerely thank the reviewer for this insightful and constructive feedback. We agree that a more dedicated and in-depth discussion of our proposed Margin-Aware Predictive Uncertainty (MPU) metric is crucial for the paper's clarity and contribution.
>
> Following this valuable suggestion, we have made a substantial revision by introducing a new dedicated subsection, **Subsection 4.4: Margin-Aware Predictive Uncertainty**, to thoroughly elaborate on the MPU. This new section is designed to directly address the points raised by the reviewer:
>
> 1.  **On MPU's Characteristics and Advantages:**
> *   We now provide an intuitive, visual comparison in the newly added **Figure 3**. This figure concretely illustrates how MPU overcomes the limitations of conventional uncertainty metrics (VoE, DE, MI) in a controlled scenario, showcasing its superior sensitivity and interpretability.
> *   Following the figure, we explicitly detail the **"Characteristics and Advantages of MPU"** in an enumerated list, highlighting three key strengths: **1) Intuitive and Interpretable**, **2) Sensitivity**, and **3) Versatility**. This provides the detailed discussion the reviewer rightfully requested.
>
> 2.  **On MPU's Deeper Connections to Preceding Theory:**
> Interestingly, MPU reveals a deep connection to two key concepts from prior literature.
> First, methods like PostN [1] have established the use of the Dirichlet strength, $S$, as a holistic measure of epistemic certainty. Second, the work of [2] highlighted the efficacy of using the concentration parameter of the predicted class, $\alpha_{\hat{y}}$, for multi-source domain adaptation.
> We find that the MPU metric elegantly unifies these two perspectives through its formulation, as defined below:
> $$
> 	\text{MPU}(\boldsymbol{\alpha}) =(K-1)\alpha_{\hat{y}} - \sum_{j \neq \hat{y}} \alpha_j= K \alpha_{\hat{y}} - S, \quad \text{where } S=\sum_{j=1}^{K}\alpha_j
> $$
> This formulation can be viewed through the lens of constrained optimization. Specifically, MPU can be interpreted as the solution to a Lagrangian dual problem that seeks to maximize the Dirichlet concentration for the predicted class $\alpha_{\hat{y}}$ while being regularized by the total Dirichlet strength $S$.
>
> We are grateful to the reviewer for guiding us to make these important improvements.
>
> ### Reference
> [1] Posterior Network: Uncertainty Estimation without OOD Samples via Density-Based Pseudo-Counts.Charpentier, Bertrand, Daniel Z\"{u}gner and G\"{u}nnemann, Stephan. NeurIPS 2020.
>
> [2] Evidential Multi-Source-Free Unsupervised Domain Adaptation. Pei, Jiangbo and Men, Aidong and Liu, Yang and Zhuang, Xiahai and Chen, Qingcha. TPAMI 2024.
>
> ### Response to weakness 2:
>
> We thank the reviewer for raising this important point. The relationship between the Optimization-Consistency (OC) property of Differential Entropy (DE) and the proposed Margin-Aware Predictive Uncertainty (MPU) indeed deserves clearer explanation.
>
> **Shared Monotonic Trend.**
> DE and MPU share a monotonic descent trend as training progresses. For DE, the OC principle guarantees that its uncertainty decreases whenever the loss decreases. For MPU, its derivation as a tight lower bound of the UCE loss ensures that it decreases in direct correspondence with loss minimization. Thus, both metrics reflect the model’s transition from uncertainty to confidence in a consistent manner.
>
> **Different Local Sensitivities.**
>
> Despite the shared monotonicity, the local curvature and gradient sensitivity of the two measures differ. DE is guaranteed to follow the correct trend but may respond more smoothly due to its entropy-based formulation. In contrast, MPU—being directly derived from the loss’s lower bound—exhibits higher sensitivity to parameter updates. Consequently, a parameter change may produce a sharp drop in the loss and MPU, while DE may change more mildly, or vice versa. This reflects the fact that these measures operate on the same global trend but on different scales of responsiveness.
>
> **Conclusion of Their Relationship.**
>
> In summary, the relationship between DE and MPU should be understood as a strong monotonic correlation rather than a linear one. DE serves as a theoretically sound uncertainty measure that satisfies OC, ensuring correct alignment with the loss. MPU goes one step further: it is the measure that is closest to the UCE loss itself and therefore provides the most loss-consistent (and thus most sensitive) uncertainty estimation under the OC principle. DE is a valid and theoretically justified measure, but MPU is the most appropriate choice for UCE-trained models due to its direct linkage to the loss landscape.
>
> **Action for Revision:**
>
> In our revised manuscript, we will add a dedicated subsection to explicitly detail this relationship. Thank you again for helping us improve the clarity of our paper.

---

> ### Author Response · Authors · 2025-11-24
> **Response to Weakness 3-4**
>
> ### Response to weakness 3:
> We thank the reviewer for this question, which gives us an opportunity to clarify our training objective.
>
> In our work, we exclusively use a single training loss, which is the UCE loss defined in Equation (1). We would like to clarify that the UCE loss is precisely the proposed "MPU-related loss" that the reviewer is asking about. The MPU uncertainty measure itself is derived from the principles of the UCE loss, and therefore, UCE loss is the sole objective function we optimize during training.
>
> To make this clearer, we have now explicitly stated this in the main text of our experimental section. We have also ensured that the caption of Table 1, where the UCE loss is mentioned, is more prominent. We hope this clarification resolves the reviewer's concern.
>
> ### Response to weakness 4:
> We thank the reviewer for this valuable suggestion to further assess the robustness of our method. We agree that evaluating performance in challenging settings like noisy labels and few-shot learning is important.
>
> **1. Regarding Noisy Settings:**
> Following the reviewer's advice, we have conducted a new set of experiments to evaluate our method's performance under label noise. The detailed setup and results are now included in **Appendix D**. We are pleased to report that our method demonstrates strong robustness against noisy input on classification accuracy and misclassification detection,
> showcasing another practical advantage of our approach.
>
> **2. Regarding Few-Shot Settings:**
> Thank you for the valuable suggestion to include few-shot learning experiments. We have carefully considered this direction. However, our paper focuses on a different core question: given a converged EDL model trained with UCE loss, which uncertainty measure is theoretically and empirically most aligned with the Optimization-Consistency principle?
>
> Few-shot scenarios typically suffer from severe data scarcity, and models trained under such settings often do not converge to a stable solution. As a result, the optimization landscape becomes highly irregular, making it difficult to meaningfully compare uncertainty measures in a way that reflects the theoretical properties we analyze (e.g., optimization-consistency, margin-aware behavior, Dirichlet strength scaling).
>
> In other words, introducing few-shot experiments would shift the focus away from our main contribution and place evaluation in a regime where the optimization behavior of UCE-trained EDL models is fundamentally unstable, making the comparison of uncertainty measures unreliable and not theoretically interpretable.
>
> To maintain a clear narrative and ensure that our empirical results faithfully reflect the theoretical principles developed in the paper, we therefore chose not to include few-shot experiments.

---

### Author Response · Authors · 2025-11-29

Dear Area Chair,

Thank you for overseeing the review process during this challenging time. We would like to briefly summarize the key status of our paper prior to the discussion suspension:

**Positive Momentum**: The reviewers with initial scores of 8, 6, and 6 remained supportive after our rebuttal.

**Critical Update**:  **Reviewer #6f5n (Initial: 2)** responded positively to our rebuttal, **raising their score to 4** and asking one additional question. We submitted a response to this inquiry, but further engagement was prevented by the system lock.

We respectfully ask you to consider the updated status (8, 6, 6, 4), noting that all constructive discussions and score revisions were conducted **entirely prior to the massive data leak**. We remain confident in the significance of our work.

Sincerely,

---

### Meta-Review · Area_Chair_JQp2 · 2026-01-06

**Summary:**

Reviewers broadly agree that the paper makes a strong theoretical contribution by reframing Evidential Deep Learning through an optimization lens and establishing a principled connection to margin-maximization. The proposed optimization-consistency principle and the Margin-aware Predictive Uncertainty (MPU) measure are viewed as well-motivated and empirically effective. While some concerns were raised about assumptions, the overall technical quality and clarity meet the ICLR bar.

**Reviewer Concerns:**

The rebuttal and follow-up discussion substantially addressed the main concerns, including clarifying the claimed “unique” property of the UCE loss, strengthening the theoretical connection to SVMs, expanding experiments (noisy, corrupted, and multimodal settings), and improving presentation and attribution. The initially critical reviewer acknowledged these clarifications and raised their score accordingly. Remaining concerns are largely about scope limitations and reliance on well-optimized models, which are now clearly stated and do not block acceptance.

**Reviewer Scores:**

Reviewer 6f5n explicitly raised their score after the rebuttal, moving from a clear reject to a borderline or neutral position. Reviewer N3WT strongly advocated for acceptance, while reviewers PQRx and WeXo would likely maintain their accept-leaning assessments. Overall, this results in a consensus leaning toward acceptance.

---

### Decision · Program_Chairs · 2026-01-26

Accept (Poster)